



# Strain response and energy dissipation of floating saline ice under cyclic compressive stress

Mingdong Wei[1], Arttu Polojärvi[1], David M. Cole[2], Malith Prasanna[1]

[1]Aalto University, School of Engineering, Department of Mechanical Engineering, P.O. Box 14100, FI-00076
Aalto, Finland.
[2]ERDC-CRREL (Ret.), 72 Lyme Rd., Hanover, NH 03768, USA

*Correspondence to*: Arttu Polojärvi (arttu.polojarvi@aalto.fi)

**Abstract.** Understanding the mechanical behavior of sea ice is the basis of ice mechanics applications. Laboratory-scale work on saline ice has often involved dry, isothermal ice specimens due to the relative ease of testing. This approach does not address the fact that the natural sea ice is practically always floating in seawater and typically has a significant temperature gradient. To address this important issue, we have developed equipment and methods for conducting compressive loading experiments on floating laboratory-prepared saline ice specimens. The present effort describes these developments and presents the results of stress-controlled cyclic compression experiments. We conducted the experiments on dry, isothermal (-10°C) ice specimens and on floating ice specimens with a naturally occurring temperature gradient. The experiments involved ice salinities of 5 and 7 ppt, cyclic stress levels ranging from 0.04–0.12 MPa to 0.08–0.25 MPa and cyclic loading frequencies of $10^{-3}$ Hz to 1 Hz. The constitutive response and energy dissipation under cyclic loading were successfully analyzed using an existing physically based constitutive model for sea ice. The results highlight the importance of testing warm and floating ice specimens and demonstrate that the experimental method proposed in this study provides a convenient and practical approach to perform laboratory experiments on floating ice.

## 1 Introduction

Climate change has led to an increased interest in polar sea areas and on ice behavior, since accurate predictions on the evolution of the ice conditions are crucial for modeling the future climate. Warming climate has also resulted in a search for more efficient marine transit routes, production of offshore wind power, industrial operations related to extraction of hydrocarbons, and even tourism in the north (Gagne et al., 2015; Serreze and Stroeve, 2015; Kern et al., 2019). Structural loads due to sea ice make these activities challenging. In-depth understanding of the physical and mechanical properties of sea ice is required to develop tools for modeling future ice conditions, the related ice–ice and ice–wave interaction problems, and for the design of safe and sustainable offshore structures (Dempsey, 2000; Feltham, 2008, Herman et al., 2019a,b; Lu et al., 2018a,b; Ranta



et al., 2017, 2018a,b; Tuhkuri and Polojärvi, 2018; Polojärvi et al., 2015; Voermans et al., 2019; Cheng et al.,

2019; Li et al., 2015).

This paper studies the mechanical behavior of laboratory-prepared saline ice specimens under cyclic loading.

This type of loading occurs in wave–ice and some ice–ice and ice–structure interaction problems – all important

in the changing polar environment. For example, warming climate causes an increase in the amount of open

water and broken ice fields, which strengthens the impact of waves on sea ice. From the perspective of pure ice

mechanics and modeling of ice, the cyclic loading experiments yield information on the elastic and viscoelastic

components of strain and their dependence on the physical, or microstructural, characteristics of the ice. The

cyclic loading tests also give insight into the fatigue of ice (Mellor and Cole, 1981; Murdza et al., 2019;

Schulson and Paul, 2009; Iliescu et al., 2017; Iliescu and Schulson, 2002).

Cyclic loading experiments on freshwater ice have been performed since the eighties (Mellor and Cole, 1981)

and on saline ice since the mid-nineties (Cole, 1995; Cole and Durell, 1995; Cole et al., 1998). Cole and Durell

(1995) studied the effects of temperature (from -5 to -50°C), cyclic stress amplitude (from 0.1 to 0.8 MPa) and

loading frequency (from $10^{-3}$ to 1 Hz) on the response of laboratory-grown saline ice; the ice response was

revealed sensitive to variations in these factors. Cole et al. (1998) investigated the response of columnar

first-year sea ice to cyclic loading and found that the elastic, anelastic, and viscous strains varied according to the

relation between the loading and preferred c-axis directions of the specimens. More recently, Heijkoop et al.

(2018) conducted strain-controlled cyclic compression tests of sea ice using cylindrical specimens having a

length of 175 mm and a diameter of 75 mm, and were able to ascertain the variation of storage and loss

compliances versus frequency. All of this earlier work has been performed using cold and dry specimens.

An often-overlooked issue in laboratory-scale experimentation is that the majority of the scenarios involving ice

deformation, such as the wave-ice interaction problems, involve ice that is floating on water. Floating ice always

has a through-thickness temperature gradient, resulting in a through-thickness gradient in its mechanical

properties. Addressing this aspect is an indispensable factor to account for in the experimentation, especially for

ice having temperatures above -10°C (Cole and Durell, 2001). The temperature gradient is implicitly taken into

account in in-situ experiments on floating ice. The cost of in-situ experimental campaigns, however, are high and

the experiments often require specially designed loading devices (Vincent and Dempsey, 1999; Dempsey et al.,

1999, 2018; Cole and Dempsey, 2004). Consequently, the relatively low costs and convenience of

laboratory-scale work motivate the development of viable methods for laboratory-scale experiments on floating

ice. Such experiments can be used, for example, for thorough validation of material models aiming to account

for the temperature gradient in ice.

In the study presented here, stress-controlled cyclic compression experiments were performed on laboratory-grown S2 saline ice specimens floating on water. For comparison, replicate experiments were performed on dry, isothermal (-10°C) specimens. The varied parameters were the ice salinity, the stress amplitude and mean stress level, and the frequency of cyclic loading. The results clearly show how the dry and the floating ice specimens exhibit significant differences in their mechanical behavior; for example, the unrelaxed moduli of floating specimens are less than half of those of dry specimens. This implies that the future laboratory-scale experimentation on ice should be more often performed using floating ice specimens instead of dry specimens. The results are comprehensively analyzed and discussed, and they are compared with the predictions obtained using a physically-based model introduced in Cole (1995). The analysis shows that the model has the capability of predicting the behavior of floating ice. On a general level, the observations here increase understanding of the behavior of sea ice in its natural conditions. Such insight on floating, rather warm, ice is important due to the future ice conditions, where on average the ice can be expected to be warmer than now (Boe et al., 2009; Blockley and Peterson, 2018; Ridley and Blockley, 2018).

The paper is organized as follows. Section 2 describes the specimen preparation, the experimental set-up, and the matrix of experimental variables. Section 3 presents the results from the experiments. Section 4 analyzes the results based on a physically-based constitutive model introduced in Cole (1995). Subsequently, Section 5 discusses our findings with references to earlier work and Section 6 gives our conclusions.

## 2 Laboratory experiments of saline ice under cyclic loading

### 2.1 Saline ice specimen preparation and characterization

The ice was grown in the cold room of Department of Mechanical Engineering, Aalto University using a tank having dimensions of 1.15 m × 1.15 m × 0.98 m (width × length × depth). The salinity of the tank water was controlled by mixing aquarium salt (Coral Pro Salt) with tap water. As one of the goals was to study the effect of ice salinity, the ice was grown from water with bulk salinities of 24 and 34 ppt.

Specimens having the nominal dimensions of 0.60 m × 0.30 m × 0.10 m were prepared by placing high-density polyethylene molds into the tank before seeding (Figure 1). The tolerance of the specimen dimensions was ±2 mm (in all stress calculations below, the measured dimensions of the specimens were used). The rectangular molds were open from the top and bottom and floated on the water with 2–3 mm of freeboard. Next, the saline water was chilled to about -1.5℃ and seeded by spraying very fine mist of fresh water into the cold air over the tank. Since the cold room temperature was about -20℃, the ice crystals were generated as the fine mist



90  encountered the cold air. Spraying was continued until a thin layer of ice crystals was formed over the entire tank

surface.

After the seeding, the room temperature was changed to a stable -14°C for three days and then -10°C for two

days for growing the ice to the desired thickness. Rigid foam insulation was placed on the sides and bottom of

the tank and heating cables were placed around the bottom perimeter to inhibit freezing from the structural

95  members of the tank. Additionally, a hose for draining excess water was installed near the base of the tank, to

prevent the accumulation of water pressure under the ice sheet during its growth. Such pressure could cause

microcracking of ice or generate additional loads on the tank. It took about five days for the ice to reach the

desired 0.10 m thickness. The ice sheets made using 24 ppt and 34 ppt-saline water reached average salinities of

5 and 7 ppt, respectively, and their densities were about 886 and 879 kg·m$^{-3}$, respectively.

The specimens used in the dry experiments (Figure 2a and 3) were sealed in plastic bags and stored in a freezer

for 1-2 days before testing. The freezer temperature was set to -10°C. However, for the floating ice experiments

(see Figure 2b), the specimens and some cold water from the growth tank were packed in the plastic bags and

then quickly transferred to the basin (located in the same cold chamber with the growth tank) where the

experiments were conducted. In this way, any major draining of the brine was minimized during the transfer.

The ice specimens were always taken out from the tank in the molds and subsequently ejected.

Before the experiments, the S2 microstructure of the ice was verified by producing and inspecting thin sections

as detailed by Langway (1958). Of special interest was the microstructure close to the specimen boundaries since

such molds are somewhat uncommon in ice specimen preparation. Figure 4 shows two typical thin sections from

near the boundary of a specimen produced using 34-ppt-saline water; one is for the vertical and the other for the

horizontal direction as indicated. Figure 4a shows the columnar structure of the ice. The horizontal view in

Figure 4b provides a way to estimate the average grain size by dividing the section area by the number of grains;

the average grain size was about 3.5 mm. To further confirm whether the ice type was S2 or S3, the universal

stage was employed to check the c-axis alignment of the grains. Figure 5 shows a Schmidt equal area net pole

projection for one of the thin sections. In the figure, the data points are uniformly scattered around the

circumference, rather than gathering around a small section of it, indicating that the c-axes of the grains are

oriented randomly in the horizontal plane, so the ice is clearly of type S2.

## 2.2 Equipment, test matrix and experimental procedure

The experiments were conducted in the same cold chamber used for the saline ice production. As Figures 2 and 3

illustrate, an externally mounted electrohydraulic cylinder applied loads to the specimens. The piston passed


through a sealed port in the side of the test tank. The piston had a maximum stroke of 800 mm and a loading capacity of 100 kN for compression and 60 kN for tension. The test basin was constructed of waterproof plywood. The system employed a camera for remote monitoring of the experiments, a three-channel temperature datalogger to record the temperature profile of floating ice during testing, a Data Acquisition Processor (DAP) board (model: Data Translation DT9834), and fixtures for arranging displacement sensors to the ice specimens. The temperature datalogger had a maximum sampling rate of one datum per five seconds, its measurement range was from -100 to 1300°C, and a resolution of ± 0.1°C. The DAP board could achieve a maximum scan rate of 500,000 samples per second.

The loading system has a self-equilibrating geometry in that the compressive force applied by the electrohydraulic piston and transmitted through the specimen is balanced by a tensile force in the external load frame (Figure 3). Care was taken to center the loading piston and the reaction plates on the vertical dimension of the load frame.

The ice deformation was measured as follows. Before the test, two 9-mm-diameter holes were drilled on the specimen with the distance of 100–150 mm from the two ends of the specimen. Two iron rods with the cross-section of 5 mm × 5 mm were then inserted into the holes and frozen in place using a small quantity of cold fresh water. The relative displacement of the rods was monitored by two linear variable differential transformer transducers (LVDTs, model: HBM WA2, with a measurement range and resolution of 2 and ± 0.001 mm, respectively). This relative displacement was used to determine the strain response of the specimen. As shown in Figures 2 and 3, to aid in the relative displacement measurement, a rectangular steel pipe was placed across the basin and attached to the LVDTs.

The through-thickness thermal gradient of the ice specimens was recorded in all wet experiments. The air temperature was kept at -10°C during all testing. To measure the gradient, three small holes with the diameter of 2.5 mm and the penetration depth of 50 mm were drilled on one side face of the ice floe, then three probes connected to a temperature datalogger were frozen into the holes. As shown in a sketch of Figure 6a, one measurement point was located in the middle of the specimen from the vertical direction, and the other two were 1.5 mm away from the upper and lower surfaces of the specimen, respectively. The temperature was recorded after the specimen floated in water for a period of time until the readings did not change on the scale of the test time. During each floating ice experiment, temperature readings from all three channels remained constant; that is, they did not vary with the change of the cycle number or the frequency of cyclic loading. This suggests that the thermal gradient inside the specimen was unaffected by the cyclic loading.



Figure 6b presents typical temperature profiles measured in two floating ice specimens. For the 5-ppt-saline floating ice, the temperature near the top surface, in the middle and near the bottom surface of the specimen was measured to be -3.0, -2.3 and -2.2°C, respectively, while the values were -3.1, -2.4 and -2.1°C, respectively, when the 7-ppt-saline ice floe was tested. These temperature readings suggest that the average temperature of the

floating ice was -2.5°C, much higher than the air temperature of the cold chamber (-10°C). Additionally, the temperature readings always showed a nonlinear relationship with depth in the specimen. The water temperature at five centimeters below the water surface was measured to be around -1.8°C in the wet tests. During the period of setting up an experiment, a thin layer of ice formed on top of the basin surface. The ice layer was manually broken before each ice experiment.

The test matrix called for applying relatively low stress cyclic compressive force with the system in load control (for example, the applied force rather than displacement is controlled). Figure 7 presents the loading waveforms used in the tests and the test matrix is given in Table 1. In the experiments, the ice was first loaded to reach an initial compressive stress state, then the cyclic loading was applied. In Figure 7a, $\sigma_{max}$ and $\sigma_{min}$ represent the upper and the lower bounds of the sinusoidal compressive stress, respectively, and $T$ denotes the period for one

loading cycle. As indicated by Table 1, six periods ranging from 1 to $10^3$ s were employed; these periods cover the main range of wave periods, which usually vary from several seconds to tens of seconds (Reistad et al., 2011; Zijlema et al., 2012). Note that in each dry test, only a fixed frequency was applied. Once the test was completed, the specimen was allowed to recover for 15 minutes before subsequent load cycles were applied. For all dry tests, the duration of the initial linear loading was fixed to be 1 s. In each of the tests corresponding to $T$ = 1, 5, 10 and

100 s, 18 cycles were applied ($N$ = 18), while for those with $T$ = 500 and 1000 s, $N$ = 9 and 4, respectively. Whether in dry or wet tests, the sequence of loading cycles was in ascending order of $T$. After the initial linear loading was applied in the wet tests, the cyclic loads with increasing periods were applied except that for $T$ = 1, 5 or 10 s, $N$ = 10, for $T$ = 100 s, $N$ =2, and for $T$ = 500 or 1000 s, $N$ = 1. The reason for this loading scheme was to shorten the time spent on the floating ice experiment and thus minimize problems with freezing of the water in

the basin.

As for the stress levels applied in the cyclic loading, the dry specimen was subjected to the stress varying from 0.08 (lower boundary of a load cycle) to 0.25 (upper boundary) MPa, which could be justifiably expected to not lead to severe damage of the specimen while being high enough to generate measurable strain (Cole and Dempsey, 2001). The floating ice experiments were conducted using stresses varying from 0.04 to 0.12 MPa.

The reason for using a smaller stress level in the wet tests was that the floating ice specimens, due to their higher temperature, may start to generate significant damage on lower stress levels than the colder, dry specimens. For





each case listed in Table 1, two specimens harvested from two ice sheets were subjected to each set of conditions. The specimens were named according to their salinity, test conditions (dry/wet), and the ice sheet from which originated from. For example, specimen name Dry-5ppt-1 indicates that the specimen came from the first

5-ppt-saline ice sheet and was tested under the dry, isothermal conditions.

Figure 7b illustrates how the energy dissipation in the cyclic loading experiments was calculated. For common engineering materials subjected to uniaxial cyclic compression, the strain along the compressive direction versus the stress is usually characterized by hysteresis loops. The energy density dissipated in a loading cycle can be determined by integrating the stress-strain curve. As shown in Figure 7b, the area under the loading curve

(region ABCF) represents the maximum strain energy input via the testing machine during a load cycle, and the area under region CDEF denotes the strain energy released during the unloading portion of the cycle. The energy density dissipation (EDD) in one full loading-unloading cycle is given by the difference between the areas of regions ABCF and CDEF (Liu et al., 2017, 2018). The energy density consumed in the hysteresis loop can, in general, be attributed to the internal friction and, in some cases, damage to the material. For each cycle, the

energy dissipation rate (EDR) can be defined as the ratio of the dissipated energy to the input energy, namely the area of the region ABCDE divided by that of the region ABCF.

### 3 Experimental results

Figure 8 presents strain-time plots obtained from the dry experiments using specimen Dry-5ppt-1. Under cyclic compressive stresses, the strain response shows a reciprocating change. Moreover, the strain-time curves

manifest the following feature: If a line would be drawn through the maximum value for each cycle, the slope of the line would first decrease from its initial value for some number of cycles and then settle down to an almost constant value. This observation means that the strain response of the ice specimen under sinusoidal compressive stress reaches a relatively steady stage after the initial transient of the anelastic strain is exhausted, and that in this steady stage the accumulated strain level increases linearly with time because the average stress is

compressive. Thus, the latter behavior is an indication of the specimen meeting a steady state in terms of viscous flow. In addition, Figure 8 shows that the longer the period $T$ of cyclic loading, the larger the strain amplitude in one steady-stage cycle.

Figure 9 shows the stress-strain curves for the same experiments. In each case, the area of the hysteresis loop for the first few cycles is comparatively large, and then it gradually decreases to a constant value as the specimen

reaches a steady deformation stage described in the previous paragraph. Thus, the EDD (Figure 7b) declines



from a relatively large value to an approximately constant value. By comparing the size of the steady-state hysteresis loops for the different frequencies, it seems that the longer the period of cyclic loading, the larger the area of the hysteresis loop; this is especially evident when one compares the hysteresis loops of the experiment having $T = 1000$ s with those corresponding to shorter periods.

Figures 10 and 11 display a set of strain-time plots and the corresponding stress-strain curves for the wet experiments on the 5-ppt-saline ice specimens, respectively. The curves in these two figures show similar features to those in Figures 8 and 9 for dry experiments: The floating ice also reached a steady state of deformation after some loading cycles and the amplitude of the steady-state strain response still increased with $T$. Similar to the results of the dry experiments, Figure 11 indicates that the longer the loading period, the larger the

strain increment of the wet specimen under one steady-state loading-unloading cycle. Comparison of Figures 9 and 11 in the steady state shows that for constant $T$ both, the strain increment per cycle and the area of one hysteresis loop, are larger in the wet than in the dry experiments. This is the case even if the stress levels are lower in the wet experiments. This observation manifests that the floating specimens have a stronger viscous flow capacity and show a more pronounced viscoelastic response than the dry specimens.

The energy density dissipation (EDD) and the energy dissipation rate (EDR) per cycle during steady-stage deformation (Section 2.2 and Figure 7b) allow quantitative comparisons of the inelastic behavior of specimens as a function of test conditions. These are presented in Tables 2 and 3 for all, except the 1-second-period, experiments, for which the hysteresis loop areas are very small in size making the measurements of energy dissipation inaccurate. Tables 2 and 3 indicate that both the EDD and the EDR decrease with the increase of

loading frequency. Moreover, under the same frequency, the 7-ppt-saline ice has larger EDD and EDR values than the 5-ppt-saline ice irrespective to the experiment type. In addition, the wet experiments always exhibit higher EDD and EDR values than the dry experiments regardless of the ice salinities. The differences in the values of EDD are especially significant for low frequencies. For example, in the experiments with $T = 1000$ s, the average value of EDD for the 5-ppt-saline dry specimens is only 24% of that of the 5-ppt-saline wet

specimens and 44% of that of the 7-ppt-saline dry specimens. However, for the experiments with $T = 5$ s, the average value of EDD in the former case is 47% and 77% of that in the latter two cases, respectively. Thus, the ice salinity and the water have a more significant influence on the energy dissipation of the ice when the cyclic loading period is long.




## 4 Material modeling

The hysteresis loops of the stress-strain curves manifest viscous and anelastic properties of the ice. According to previous studies (Cole, 1995; Leclair et al., 1999), in low-stress cyclic loading experiments, the microstructure of the ice remains unaffected by loading, or in other words, no damage occurs within the material. In the present case of polycrystalline ice, the anelastic deformation is mainly attributed to two relaxation mechanisms, lattice dislocation relaxation and grain-boundary sliding. Viscous straining is attributable to basal dislocation glide. In

this section, a dislocation-based model (Cole, 1995; Cole and Durell, 2001), which accounts for these mechanisms, is used to predict the strain response of the ice specimens based on their physical properties and experimental conditions. Here the model is only briefly described, but a detailed description can be found from Cole et al. (1998), where the model is also demonstrated to reproduce the viscous and anelastic behavior of dry specimens. Here the applicability of the model in predicting the behavior of floating laboratory-prepared ice

specimens subjected to cyclic loading is tested for the first time.

### 4.1 Brief description of the model

In the physically based model by Cole (1995) and Cole and Durell (2001), the axial strain, $\varepsilon$, of ice under uniaxial cyclic compression is considered to be composed of elastic, anelastic (delayed elastic) and viscous

components, denoted here as $\varepsilon_e$, $\varepsilon_a$ and $\varepsilon_v$, respectively. The axial strain is expressed as

$$\varepsilon = \varepsilon_e + \varepsilon_a + \varepsilon_v \cdot \tag{1}$$

In the case of sinusoidal stress waveform, $\varepsilon_e$ can be written as

$$\varepsilon_e = \frac{\sigma(\omega,t)}{E_0}, \tag{2}$$

where $\omega$ is the angular frequency of the stress waveform and $E_0$ is the unrelaxed modulus. Although detailed

expressions have been developed for the effective elastic modulus as a function of crystallography, brine and gas porosity, and temperature, a simplified approach is adopted in the present effort. The anelastic component $\varepsilon_a$ incorporates both above-mentioned relaxation mechanisms to represent the time-dependent recoverable deformation. For the steady-stage deformation, $\varepsilon_a$ of ice subjected to sinusoidal compressive stress can be decomposed as (Cole and Dempsey 2001)

$$\varepsilon_a = \sigma(\omega,t)\left[ D_1^d(\omega) + D_2^d(\omega) + D_1^{gb}(\omega) + D_2^{gb}(\omega) \right], \tag{3}$$

where the compliance terms, $D$, with the superscripts "d" and "gb" denote the compliances induced by dislocation and grain boundary sliding, respectively. The compliance terms are defined as (Cole et al., 1998)



$$D_1^{\mathrm{d}}\left(\omega\right) = \delta D^{\mathrm{d}}\left\{1 - \frac{2}{\pi}\tan^{-1}\left[e^{\left(\alpha^{\mathrm{d}}s^{\mathrm{d}}\right)}\right]\right\} \tag{4}$$

$$D_1^{\mathrm{gb}}\left(\omega\right) = \delta D^{\mathrm{gb}}\left\{1 - \frac{2}{\pi}\tan^{-1}\left[e^{\left(\alpha^{\mathrm{gb}}s^{\mathrm{gb}}\right)}\right]\right\} \tag{5}$$

$$D_2^{\mathrm{d}}\left(\omega\right) = \alpha^{\mathrm{d}}\cdot\delta D^{\mathrm{d}}\frac{1}{e^{\left(\alpha^{\mathrm{d}}s^{\mathrm{d}}\right)}+e^{\left(-\alpha^{\mathrm{d}}s^{\mathrm{d}}\right)}} \tag{6}$$

$$D_2^{\mathrm{gb}}\left(\omega\right) = \alpha^{\mathrm{gb}}\cdot\delta D^{\mathrm{gb}}\frac{1}{e^{\left(\alpha^{\mathrm{gb}}s^{\mathrm{gb}}\right)}+e^{\left(-\alpha^{\mathrm{gb}}s^{\mathrm{gb}}\right)}}\,, \tag{7}$$

where $S^{\mathrm{d}} = \ln\left(\tau^{\mathrm{d}}\omega\right)$; $\tau^{\mathrm{d}}$ is the central relaxation time (Cole and Durell, 1995). $\alpha$ is a so-called peak broadening term, which accounts for the effect of a distribution in relaxation times of the basal plane dislocations. The grain boundary relaxation is calculated using similar mathematic expressions as the dislocation relaxation, but has a

different strength, activation energy and peak-broadening term. The activation energy is 1.32 eV for the grain boundary relaxation and 0.55 eV for the dislocation relaxation (Cole and Dempsey, 2001). The peak-broadening terms for lattice dislocation relaxation and grain-boundary sliding relaxation are typically $\approx$ 0.5 and 0.6, respectively, determined experimentally by Cole (1995), Cole and Dempsey (2001) and also validated by Heijkoop et al. (2018). The strength of the dislocation relaxation is calculated from

$$\delta D^{\mathrm{d}} = \frac{\rho\Omega b^2}{K}\,, \tag{8}$$

where $b$ represents Burgers vector ($b = 4.52 \times 10^{-10}$ m); $\rho$ denotes the mobile dislocation density, often found to be on the order of $10^9$ m$^{-2}$; $\Omega$ is an orientation factor, determining the average basal plane shear stress induced by the background normal stress ($\Omega = 1/\pi \approx 0.32$ for a horizontal specimen made of unaligned S2 ice); $K$ is a restoring stress constant, determined as 0.07 kPa for polycrystalline ice in experiments (Cole and Durell 2001).

In Eq. (1), the viscous strain $\varepsilon_{\mathrm{v}}$ is often estimated with the following formulae (Cole and Durell, 2001):

$$\varepsilon_{\mathrm{v}} = \int_0^t \dot{\varepsilon}_{\mathrm{v}}d\bar{t} \tag{9}$$

$$\dot{\varepsilon}_{\mathrm{v}} = \frac{\beta\rho\Omega^{1.5}b^2\sigma_{\mathrm{creep}}}{B_0}e^{\left(-\frac{Q_{\mathrm{glide}}}{kT^*}\right)}\,, \tag{10}$$

where $\beta = 0.3$, $Q_{\mathrm{glide}} = 0.55$ eV, and $B_0 = 1.205 \times 10^{-9}$ Pa·s. $k$ is Boltzmann's constant, $T^*$ is the temperature in Kelvins. By using the above definitions and equations, the strain of the ice specimen can be finally determined

via Eq. (1).

**4.2 Modeling**



In this study, the key quantity to be determined from the experiments is the dislocation density. With knowledge of the microstructure and orientation factor ($\Omega$), and given the experimental conditions, the anelastic term $\delta D^{\mathrm{d}}$

and the viscous strain rate can be calculated directly. Since there were insufficient data to determine $\delta D^{\mathrm{gb}}$ directly for our specimens, a value was determined empirically and found to be in line with previous findings. Although a detailed model for the elastic modulus $E_0$ is available (Cole et al., 1998), on-going work indicates that the elastic modulus of floating sea ice is substantially lower than found in isothermal sea ice under dry laboratory conditions. Pending publication of those findings, we opt here to determine $E_0$ empirically as well.

This was done by trial and error until the stress-strain curves generated by the model matched with those measured in the experiments with $T = 10$, 100 and 500 s. The values determined for the parameters are tabulated in Table 4. Then, the model based on these parameter values was applied to predict the test results for other loading periods. An example of the comparison of experimental and modeling results is presented in Figure 12, in which the steady-state strain curves from all dry experiments on the 5-ppt-saline ice specimen are

accompanied by the simulated ones.

Figure 12 shows that the modeled strain records compare well with those from the experiments with period $T = 1$, 5 and 1000 s. The model reproduced the steady-state strain response of the specimens very well for all tested frequencies. Figure 13 presents the stress-strain hysteresis loops from the same experiments together with those produced by the model. The hysteresis loops generated by using the model are very similar to those from the

experiments. The loop area increases with $T$ in both the experiments and simulations.

For assessing the model in more detail, the values of EDD and EDR derived using it are given in Tables 5 and 6, respectively, and moreover, compared with the data from all the experiments (the test matrix is given in Table 1). Tables 5 and 6 indicate that the model predicts well the values of EDD and EDR for all cyclic loading periods studied here. In general, the error in the predicted EDD and EDR values are within 20%, with the exception of

only few cases. No obvious trend between the magnitude of error and loading period or experiment type was observed. Note that for a given ice specimen, one dislocation density value adequately modes the steady-state strain responses and energy dissipation values in tests conducted with different frequencies. The value of dislocation density, thus, remained constant through the cyclic loading.

**4.3 Further validation**

The above results show that the model can yield satisfactory predictions on the viscous and anelastic behavior of both dry and wet specimens. One may argue that the good agreement between the model predictions and the experimental results of this study only indicates the capability of the model to predict the results for the



experiments with the same stress levels as those used to calibrate the model parameters. To check whether the

model can predict the results of the experiments with different stress levels than those above, additional experiments were performed on specimen Dry-5ppt-1 with higher stress (0.1–0.3 MPa) and on specimen Wet-7ppt-1 with lower stress (0.005–0.085 MPa). The model was then used to solve the strain response and energy dissipation in these tests using the parameterization based on the experiments of Section 3 (Table 4).

Figure 14 compares the stress-strain hysteresis loops obtained in the supplementary experiments with the

predictions yielded from the model. Good agreement is observed between the predicted and measured hysteresis loops. Again, the relative errors in the EDD and EDR values are found to be less than 20% for the vast majority of data sets. The model is, indeed, capable of predicting the deformation of saline ice in a dry environment or when floating in water irrespective of the stress level (given that the stress levels are moderate and do not cause an increase in dislocation density during deformation). The applicability of the model is highlighted by the fact

that once the model parameters are calibrated through benchmark experiments, it yields sound predictions for the experiments conducted with other stress levels.

**5 Discussion**

Although past laboratory-scale work has provided insight into the mechanical behavior of sea ice, the work has been mostly performed using relatively cold, dry and isothermal specimens. The results above indicate that more

attention should be paid to the mechanical behavior of relatively warm, floating ice. Here efforts were made to develop a relatively low-cost, convenient and useful approach for laboratory-scale floating ice experiments and for preparing small-scale, saline columnar ice specimens. The thin sections (Figures 4 and 5) indicated that the measures taken on the ice production guaranteed the generation of S2 ice. Moreover, the thin sections showed that the molds used for specimen preparation do not influence the columnar ice microstructure. This is

encouraging, as molds are not often used when preparing S2 ice specimens, but they proved to significantly simplify the preparation of test specimens with accurate dimensions. The molds are especially effective in the experiments on floating ice specimens: The specimens thus produced have proper dimensions without the need for sawing or other procedures, have a realistic temperature gradient and brine loss is minimized.

The results above are in qualitative agreement with what would be expected for floating ice, as for example,

floating specimens exhibit lower modulus and more pronounced inelastic deformation in comparison with dry specimens. This gives confidence on the methods employed in the production and mechanical testing of the specimens. Further, the good agreement between the results for floating ice and the corresponding model



predictions show quantitative validity of the approach taken here. All-in-all, the results suggest that the chosen experimental techniques (including the loading system, the strain measurement scheme, the data acquisition

system and settings) worked well and provided useful results for the analysis. An exception is some of the 1-second-period experiments, which yielded strain responses with unexpected features; this is because the hysteresis loop areas are very small in size, making accurate measurements with the used set-up challenging. This could be circumvented in the future experiments by using techniques and devices with even higher precision and setting a higher sampling rate of data acquisition. The methods proposed in this study to conduct

laboratory experiments on floating ice experiments are practical and can provide a convenient approach for relatively low-cost experimentation on floating ice.

The laboratory work here not only demonstrates the availability of the proposed experimental methods, but also contributes to the understanding of the constitutive behavior of ice. A common trend in developing material models is to ensure they have a solid physical basis, that is, that they are based on an understanding of the

physical processes that underlie the mechanical phenomena of interest. Above we used one such physically-based model introduced by Cole (1995). Earlier work, which has been based on the experiments resembling the dry experiments here, has shown that the model is capable of predicting the inelastic deformation of sea ice via dislocation-based mechanisms and is able to estimate the effective dislocation density in ice. Here the model was successfully validated against the results from the wet experiments with floating ice. Moreover,

the results indicate that once the constant dislocation density value of ice is determined, the model can predict the steady-stage deformation for the cyclic loading experiments conducted with different frequencies and stress levels. These bring confidence to the model and clearly demonstrate its usability in modeling full-scale practical applications involving floating ice, especially considering that some research has been launched to devote to a numerical implementation of the model (O'Connor et al., 2020).

The good agreement between the experimental and modeling results also motivates discussion on the parameter values in Table 4. Expectedly, the floating ice specimens have lower unrelaxed moduli than the dry specimens, regardless of the ice salinity. For the 5-ppt and 7-ppt-saline ice, the average elastic modulus of the floating specimens is 66% and 55% lower than that of the dry specimens, respectively. In the dry experiments, the isothermal 7-ppt-saline ice has a 31% lower average elastic modulus than the 5-ppt-saline ice. In the wet

experiments, the 7-ppt-saline specimen still has a lower average elastic modulus than the 5-ppt-saline specimen, but the difference, 8%, is not as prominent as in the dry experiments. Thus, the water and the related through-thickness temperature gradient have a larger influence on the elastic modulus of ice than the variation of the ice salinities studied here.



Table 4 also indicates that the dislocation densities determined for the ice specimens are on the order of

$\sim10^8$–$10^{10}$ m$^{-2}$, and thus are in good agreement with values in the $10^9$ m$^{-2}$ order of magnitude given by Cole (1995). Moreover, the dislocation density of the wet ice is approximately one order of magnitude greater than that of the dry specimen. Note that there is precedent showing the dislocation density of ice increasing with temperature (Cole and Durell 2001). The same trend is seen in this study when the experiments go from dry specimens (-10°C) to relatively warm, floating ice. In addition, the previously cited work showed that the

dislocation density increases with salinity in undeformed ice. Therefore, the calculated dislocation densities make sense physically and are generally in line with expectations.

Specimens originating from two different ice sheets were used in the experiments. Table 4 shows that for a given set of conditions, the elastic modulus of the specimens shows only small variation from one sheet to another. Compared with the elastic modulus, the relative change in dislocation density is fairly large, which may be due

to heterogeneity of ice. A similar degree of change in dislocation density was also reported by Cole and Durell (2001) related to dry, isothermal, ice specimens. As for the strength of grain boundary, its value is always 2 ($\pm$1) $\times10^{-10}$ Pa$^{-1}$ for all specimens, not showing orders of magnitude change. This value remaining fairly constant owing to very small variation in grain size between the ice specimens and the generated sheets. Thus, here the main quantity to be determined from the experimental results for studying anelastic strain response of the

specimens was the dislocation density.

The present study demonstrates significant differences in the mechanical behavior of ice in dry and wet experiments, that is, in commonly performed experiments on cold and dry specimens and in experiments on ice specimens floating in water as in nature. In both cases, the air temperature was controlled to be -10°C, and the thermal distributions inside the specimens were stable. As already shown by Cole and Durell (2001), the

temperature is a critical factor influencing the mechanical behavior of ice under cyclic loading. There are also many references showing the effect of temperature on the mechanical and physical behaviors of saline ice (Golden et al., 2007). The effect of water on the mechanical behavior of floating ice is largely due to the effect of temperature (Golden et al., 2007). Moreover, the specimens harvested from floating ice sheets lose brine once removed from the sheet; this operation may permanently alter their response. In addition, some remaining brine

(for example, some of those in capillary brine channels) must freeze during the storage process of dry specimens; this may as well lead to some difference in the macroscopic mechanical behavior (for example, in elastic modulus) of dry and wet specimens (Marchenko and Lishman, 2017; Eicken, 1992; Jones et al., 2012; Gough et al., 2012).



**6 Conclusions**

In this work, laboratory experiments were conducted to study the strain response of saline ice under cyclic compressive stresses. By using a newly proposed experimental method, the saline ice specimens were tested under dry, isothermal (-10°C) conditions or floating in saline water with a through-thickness temperature gradient. In this way, the mechanical behavior of the dry and wet specimens can be compared. Moreover, these initial experiments allow assessing the practicality of the proposed experimental method in conducting floating

ice experiments. In addition, the experimental results compared favorably with the theoretical predictions obtained using a physically based constitutive model for saline ice. Some conclusions can be drawn as follows:

1. The methods proposed in this study for making S2 saline ice and conducting laboratory experiments of floating ice provide convenient and useful approaches for relatively low-cost, indoor floating ice experiments.

2. Compared with the dry specimens, the wet specimens can consume more strain energy, deform more significantly in a single loading cycle and show stronger viscous flow capacities. One of the essential reasons is due to the through-thickness thermal gradient of the floating ice. In the environment where the air temperature is -10°C and temperature in the water column was -1.8°C, the average temperature of the saline ice was -2.5°C.

3. Since the deformation response behavior of wet and dry specimens differ considerably, laboratory experiments on saline ice should more often be performed using floating ice specimens instead of dry, isothermal specimens.

4. The dislocation mechanics of the model employed in the analysis can reproduce very well the strain response and energy dissipation of saline ice subjected to cyclic loading for floating ice or dry specimens,

and for the observed ice salinities. The results show that the prediction errors of the energy density dissipation and the energy release rate are within $\approx 20\%$.

5. For either dry or wet specimens, the higher the salinity of ice, the lower the modulus ($E_0$) and the larger the dislocation density ($\rho$). In addition, $E_0$ is much higher and $\rho$ is far smaller for the dry specimens than for the wet specimens, provided other experimental variables are consistent.

All-in-all, this study clearly suggests that more attention be paid to the differences between the mechanical behavior of dry isothermal specimens vs. wet floating specimens of saline ice. This distinction is especially important because the future applications of sea ice mechanics focus largely on relatively warm, floating ice conditions as a consequence of climate change.



**Financial support:** The authors are grateful for the financial support from the Academy of Finland through the
project (309830) Ice Block Breakage: Experiments and Simulations (ICEBES).

**Acknowledgments:** The authors thank the help from the technical staff of Aalto University Department of
Mechanical Engineering, and especially, from Kari Kantola and Veijo Laukkanen.

**Author contributions:** AP, MW and DMC designed the study. MW and MP performed the experiments. MW, AP
and DMC contributed to the interpretation of the results. MW, AP and DMC drafted the paper. All authors
commented on the text.

**Code and data availability:** The code used for material modeling is written in Matlab. Scripts used for analysis
and more detailed information of the experimental results are available from the authors upon request.

**Competing interests:** The authors declare that they have no conflict of interest.

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





Table 1 The test matrix of this experimental campaign. The campaign included two test types, two ice salinities, five periods and two test-type-dependent load levels. For each case, two specimens harvested from two ice sheets were tested.

| Case | Specimen no. | Ice salinity (ppt) | Dry/wet | Average ice temperature (°C) | Period (s) | Cyclic compressive Stresses (MPa) |
|---|---|---|---|---|---|---|
| I | Dry-5ppt-1, Dry-5ppt-2 | 5 | Dry | -10 | 1, 5, 10, 100, 500, 1000 | 0.08–0.25 |
| II | Dry-7ppt-1, Dry-7ppt-2 | 7 | Dry | -10 | 1, 5, 10, 100, 500, 1000 | 0.08–0.25 |
| III | Wet-5ppt-1, Wet-5ppt-2 | 5 | Wet | -2.5 | 1, 5, 10, 100, 500, 1000 | 0.04–0.12 |
| IV | Wet-7ppt-1, Wet-7ppt-2 | 7 | Wet | -2.5 | 1, 5, 10, 100, 500, 1000 | 0.04–0.12 |

Table 2 The energy density (J·m$^{-3}$) dissipated in a typical loading-unloading cycle of each experiment.

| Specimen no. | Frequency (Hz) | | | | |
|---|---|---|---|---|---|
| | 0.001 | 0.002 | 0.01 | 0.1 | 0.2 |
| Dry-5ppt-1 | 5.36 | 2.93 | 1.57 | 0.54 | 0.36 |
| Dry-5ppt-2 | 8.83 | 5.44 | 2.01 | 0.67 | 0.71 |
| Dry-7ppt-1 | 14.4 | 7.16 | 2.87 | 0.91 | 0.66 |
| Dry-7ppt-2 | 20.8 | 14.0 | 7.80 | 2.49 | 2.05 |
| Wet-5ppt-1 | 25.0 | 15.4 | 4.78 | 0.85 | 0.63 |
| Wet-5ppt-2 | 28.4 | 16.2 | 4.79 | 0.91 | 0.54 |
| Wet-7ppt-1 | 151 | 67.7 | 20.2 | 3.85 | 2.52 |
| Wet-7ppt-2 | 63.6 | 38.3 | 11.7 | 2.40 | 1.63 |

Table 3 The energy dissipation rate (%) in a typical loading-unloading cycle of each experiment.

| Specimen no. | Frequency (Hz) | | | | |
|---|---|---|---|---|---|
| | 0.001 | 0.002 | 0.01 | 0.1 | 0.2 |
| Dry-5ppt-1 | 35.4 | 22.7 | 17.6 | 8.68 | 5.50 |
| Dry-5ppt-2 | 48.3 | 40.2 | 22.0 | 9.86 | 9.93 |
| Dry-7ppt-1 | 55.6 | 37.3 | 21.6 | 11.0 | 8.60 |
| Dry-7ppt-2 | 55.7 | 44.7 | 32.3 | 16.0 | 14.3 |
| Wet-5ppt-1 | 89.6 | 76.1 | 44.2 | 15.5 | 11.8 |
| Wet-5ppt-2 | 87.7 | 69.8 | 39.4 | 17.4 | 11.3 |
| Wet-7ppt-1 | 98.3 | 91.7 | 70.5 | 32.0 | 25.1 |
| Wet-7ppt-2 | 88.1 | 76.0 | 56.4 | 26.3 | 20.2 |






**Table 4. Values of the model parameters calibrated for simulating the strain response of the ice specimens (in Section 5, these values are discussed and compared to those reported in references).**

| Specimen no. | Elastic modulus $E_0$ (GPa) | Dislocation density $\rho$ (m$^{-2}$) | Strength of dislocation relaxation $\delta D^d$ (Pa$^{-1}$) | Strength of grain boundary $\delta D^{gb}$ (Pa$^{-1}$) |
|---|---|---|---|---|
| Dry-5ppt-1 | 5.6 | $4.31 \times 10^8$ | $4 \times 10^{-10}$ | $1 \times 10^{-10}$ |
| Dry-5ppt-2 | 6.0 | $7.53 \times 10^8$ | $7 \times 10^{-10}$ | $1 \times 10^{-10}$ |
| Dry-7ppt-1 | 4.0 | $1.18 \times 10^9$ | $1.1 \times 10^{-9}$ | $1 \times 10^{-10}$ |
| Dry-7ppt-2 | 4.0 | $1.83 \times 10^9$ | $1.7 \times 10^{-9}$ | $3 \times 10^{-10}$ |
| Wet-5ppt-1 | 2.0 | $5.92 \times 10^9$ | $5.5 \times 10^{-9}$ | $3 \times 10^{-10}$ |
| Wet-5ppt-2 | 1.9 | $6.46 \times 10^9$ | $6 \times 10^{-9}$ | $1 \times 10^{-10}$ |
| Wet-7ppt-1 | 1.9 | $2.58 \times 10^{10}$ | $2.4 \times 10^{-8}$ | $3 \times 10^{-10}$ |
| Wet-7ppt-2 | 1.7 | $1.40 \times 10^{10}$ | $1.3 \times 10^{-8}$ | $2 \times 10^{-10}$ |

**Table 5. Modeling results of the strain energy density (J·m$^{-3}$) dissipated per loading-unloading cycle (the values given in parentheses are error percentages of model predictions relative to experimental results).**

| Specimen no. | Frequency | | | | |
|---|---|---|---|---|---|
| | 0.001 Hz | 0.002 Hz | 0.01 Hz | 0.1 Hz | 0.2 Hz |
| Dry-5ppt-1 | 5.22 (-3%) | 3.10 (6%) | 1.15 (-27%) | 0.55 (2%) | 0.33 (-8%) |
| Dry-5ppt-2 | 9.12 (7%) | 5.16 (9%) | 1.74 (-10%) | 0.69 (-9%) | 0.68 (-21%) |
| Dry-7ppt-1 | 15.1 (5%) | 8.76 (22%) | 2.65 (-8%) | 0.86 (-5%) | 0.79 (20%) |
| Dry-7ppt-2 | 21.9 (-1%) | 12.5 (-4%) | 6.44 (-9%) | 2.00 (-22%) | 1.97 (12%) |
| Wet-5ppt-1 | 27.0 (8%) | 14.9 (-3%) | 4.05 (-15%) | 0.91 (7%) | 0.73 (16%) |
| Wet-5ppt-2 | 30.3 (3%) | 16.8 (1%) | 4.58 (-6%) | 0.89 (-17%) | 0.61 (-8%) |
| Wet-7ppt-1 | 125 (-17%) | 68.5 (1%) | 18.6 (-8%) | 3.79 (-2%) | 2.58 (2%) |
| Wet-7ppt-2 | 65.4 (-17%) | 36.4 (-4%) | 10.0 (-21%) | 2.18 (10%) | 1.73 (13%) |







**Table 6. Modeling results of the strain energy dissipation rate (%) per loading-unloading cycle (the values given in parentheses are error percentages of model predictions relative to experimental results).**

| Specimen no. | Frequency | | | | |
|---|---|---|---|---|---|
| | 0.001 Hz | 0.002 Hz | 0.01 Hz | 0.1 Hz | 0.2 Hz |
| Dry-5ppt-1 | 37.4 (16%) | 25.9 (14%) | 13.9 (-21%) | 7.5 (-5%) | 5.70 (12%) |
| Dry-5ppt-2 | 49.2 (4%) | 36.7 (-1%) | 18.5 (-13%) | 9.43 (-12%) | 9.81 (-16%) |
| Dry-7ppt-1 | 53.1 (-1%) | 41.3 (-2%) | 20.2 (10%) | 8.99 (-18%) | 8.50 (12%) |
| Dry-7ppt-2 | 51.3 (-1%) | 38.9 (-2%) | 25.7 (10%) | 12.6 (-18%) | 13.9 (12%) |
| Wet-5ppt-1 | 87.8 (-5%) | 77.1 (-4%) | 40.0 (-15%) | 16.0 (3%) | 14.0 (3%) |
| Wet-5ppt-2 | 91.1 (-1%) | 82.0 (6%) | 45.1 (-4%) | 17.6 (9%) | 13.4 (26%) |
| Wet-7ppt-1 | 86.4 (-1%) | 77.4 (-3%) | 59.9 (-10%) | 34.0 (13%) | 28.2 (17%) |
| Wet-7ppt-2 | 94.5 (-1%) | 88.3 (-3%) | 52.9 (-10%) | 23.8 (13%) | 20.5 (17%) |


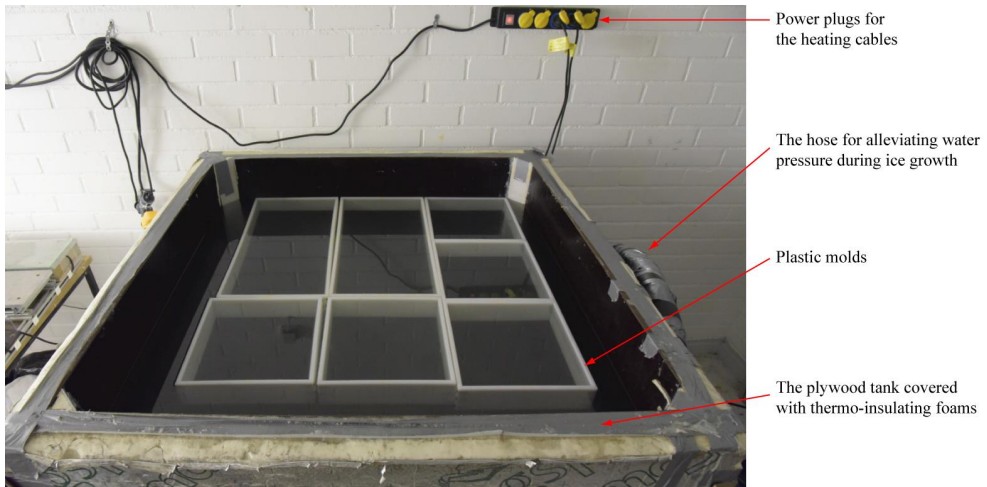

Figure 1. The tank made of plywood for growing ice specimens.


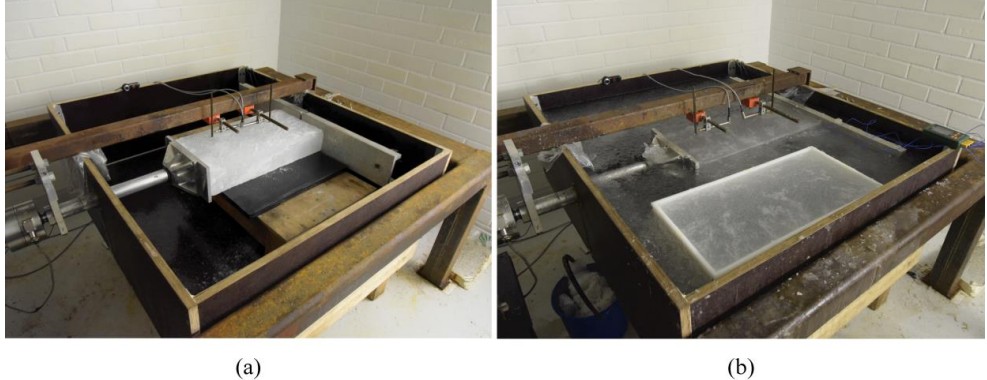

|  (a)  |  (b)  |

**Figure 2. Equipment used in the (a) dry and (b) wet experiments. One of the plastic molds used when growing ice is shown in (b). The thin ice cover of the basin, seen in (b), was broken before performing the experiments.**






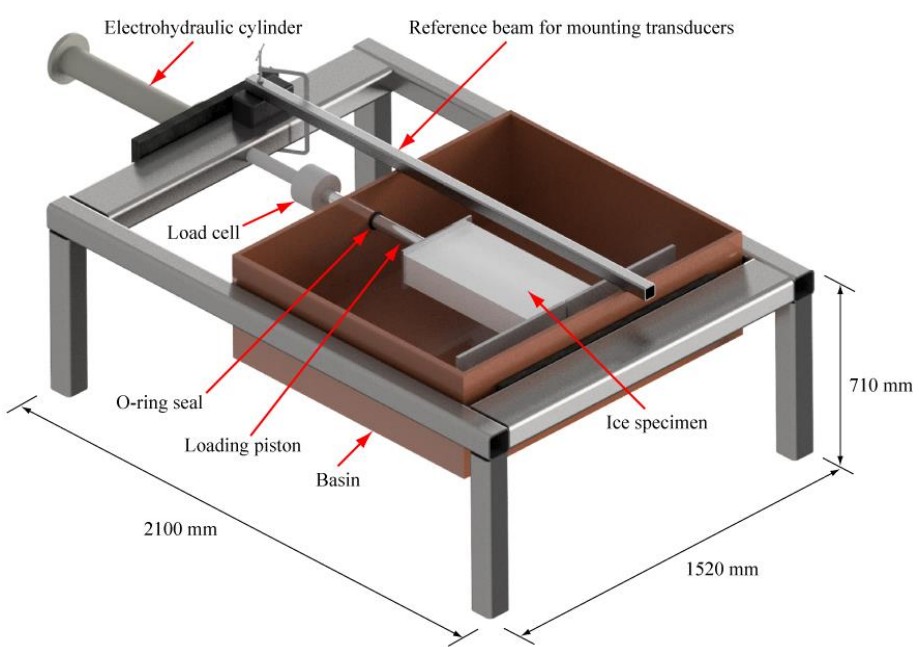

**Figure 3. A sketch of the test rig used in the experiments. The inner dimensions of the basin are 1320 mm × 1280 mm × 400 mm.**

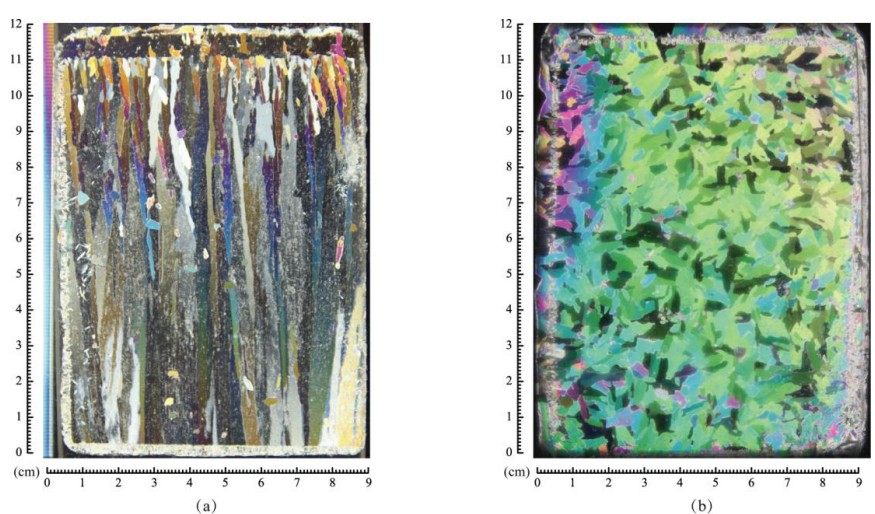


**Figure 4. One set of thin sections of the ice grown in this experimental campaign: (a) vertical; (b) horizontal.**




(Schmidt equal area net pole projection figure)

**Figure 5. A typical Schmidt equal area net pole projection drawn on the basis of grain orientations.**

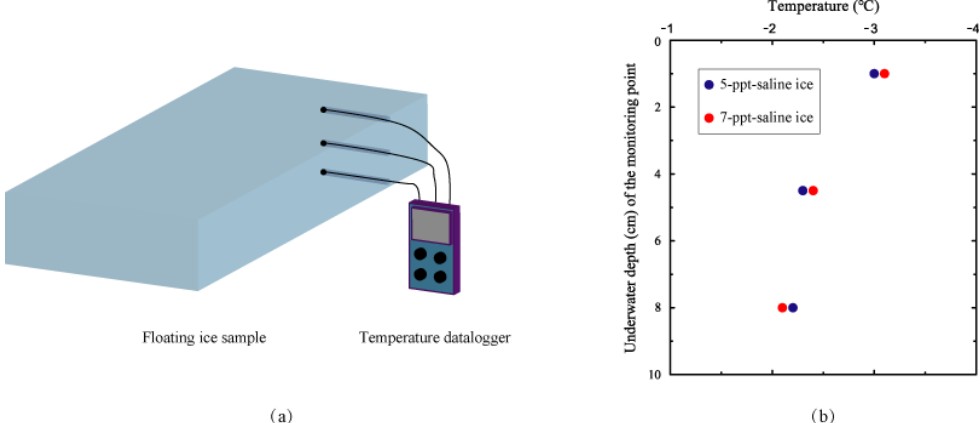

(a)                              (b)

**Figure 6. Monitoring of the through-thickness thermal gradient of floating ice: (a) a schematic diagram of the arrangement of the temperature probes and (b) the measured through-thickness thermal gradient inside two floating ice specimens.**


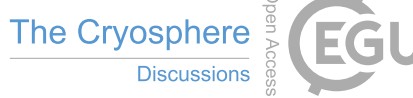

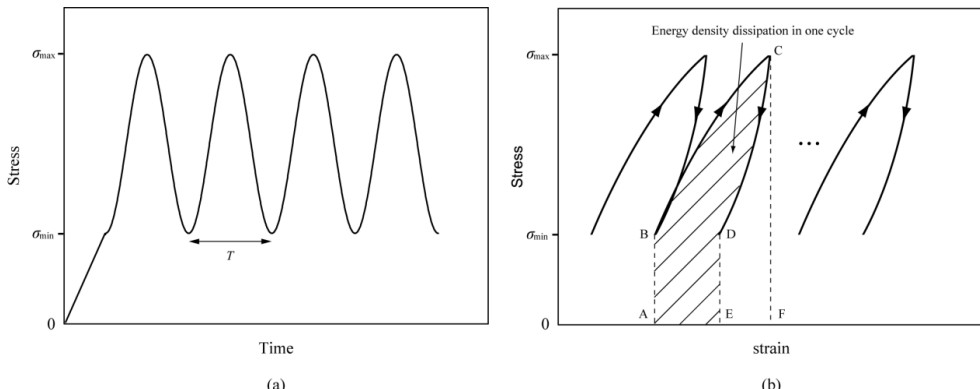

**Figure 7. Schematic diagrams of the cyclic loads: (a) the inputted stress waveform, and (b) the method for calculating the energy density dissipated in one loading cycle (in other words, region ABCDE).**


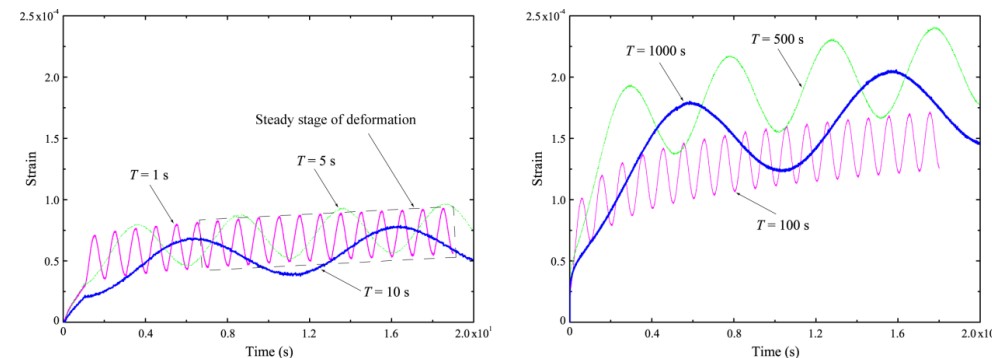

**Figure 8. Strain-time curves of specimen Dry-5ppt-1 tested with stresses varying from 0.08 to 0.25 MPa and with the temperature of -10°C.**




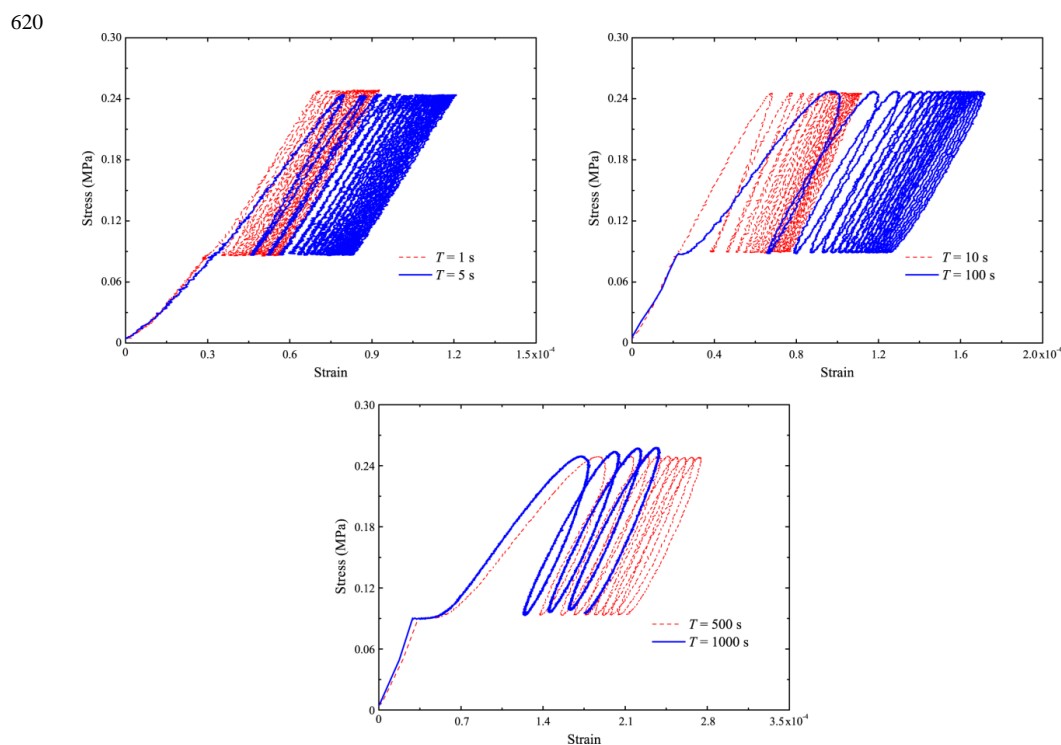

**Figure 9. Stress-strain curves of specimen Dry-5ppt-1 tested with stresses varying from 0.08 to 0.25 MPa and with the temperature of -10°C.**

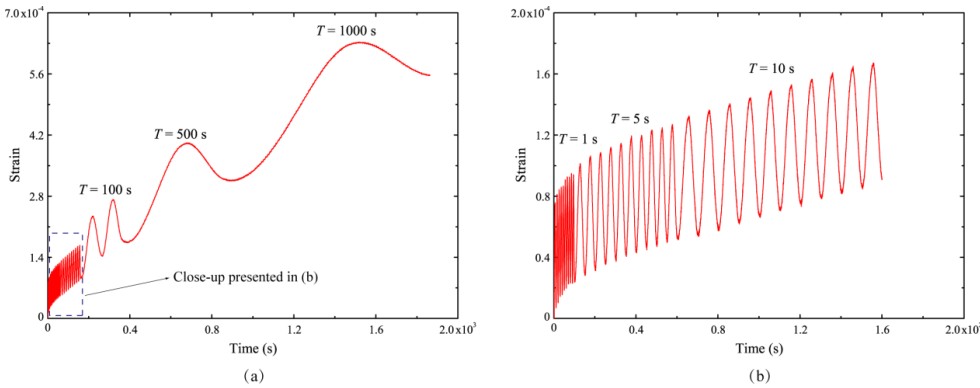


**Figure 10. Strain response of specimen Wet-5ppt-1 tested with stresses varying from 0.04 to 0.12 MPa and with the average temperature of -2.5°C: (a) shows the response for all periods, *T*, of cyclic loading, while (b) presents a close-up showing the cycles for *T* = 1, 5 and 10 s.**





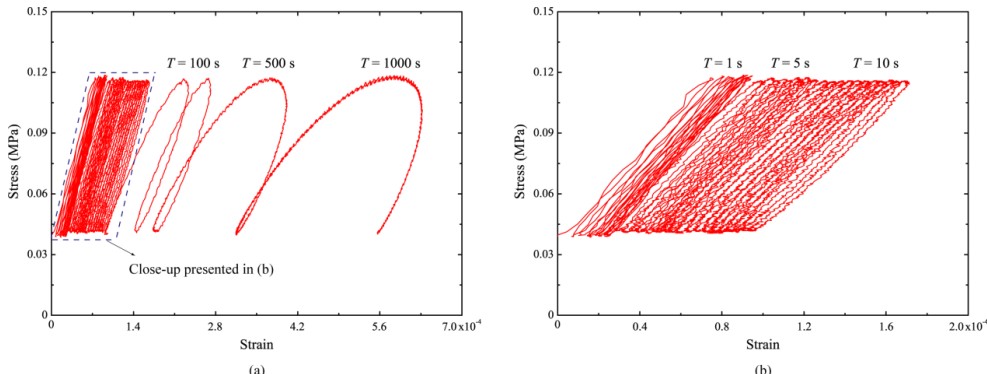


**Figure 11. Stress-strain plots of specimen Wet-5ppt-1 tested with stresses varying from 0.04 to 0.12 MPa and with the average temperature of -2.5°C: (a) shows the response for all periods, *T*, of cyclic loading, while (b) presents a close-up showing the cycles for *T* = 1, 5 and 10 s.**







**Figure 12.** Comparison between the experimentally measured strain-time curves (in the steady stage) and the results yielded by the physically based model for specimen Dry-5ppt-1 tested with stresses varying from 0.08 to 0.25 MPa and with the temperature of -10°C.





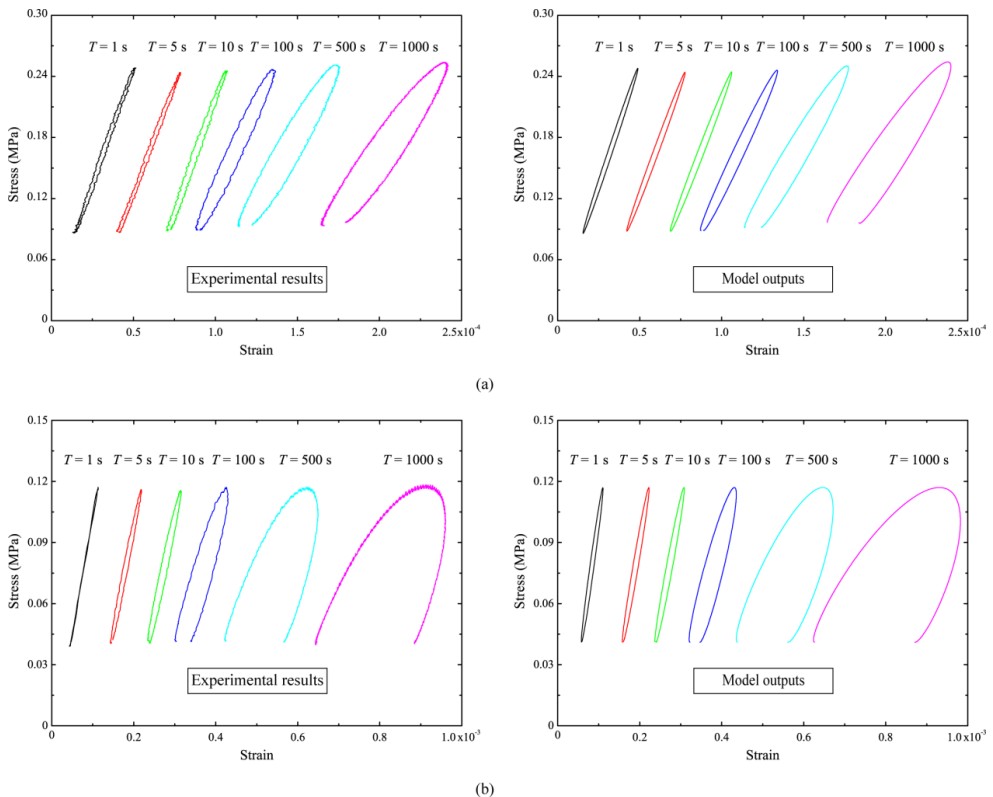

(a)

(b)

**Figure 13. Comparison between the steady-state stress-strain hysteresis loop measured in the experiments and that outputted from the model for (a) specimen Dry-5ppt-1 with the temperature of -10℃ and (b) specimen Wet-5ppt-1 with the average temperature of -2.5℃ (note that, to compare the representative hysteresis loops in all the different-frequency experiments with those outputted by the model in a more concise and intuitive way, the pre-strains before the hysteresis loops drawn here are not equal to the experimental values).**







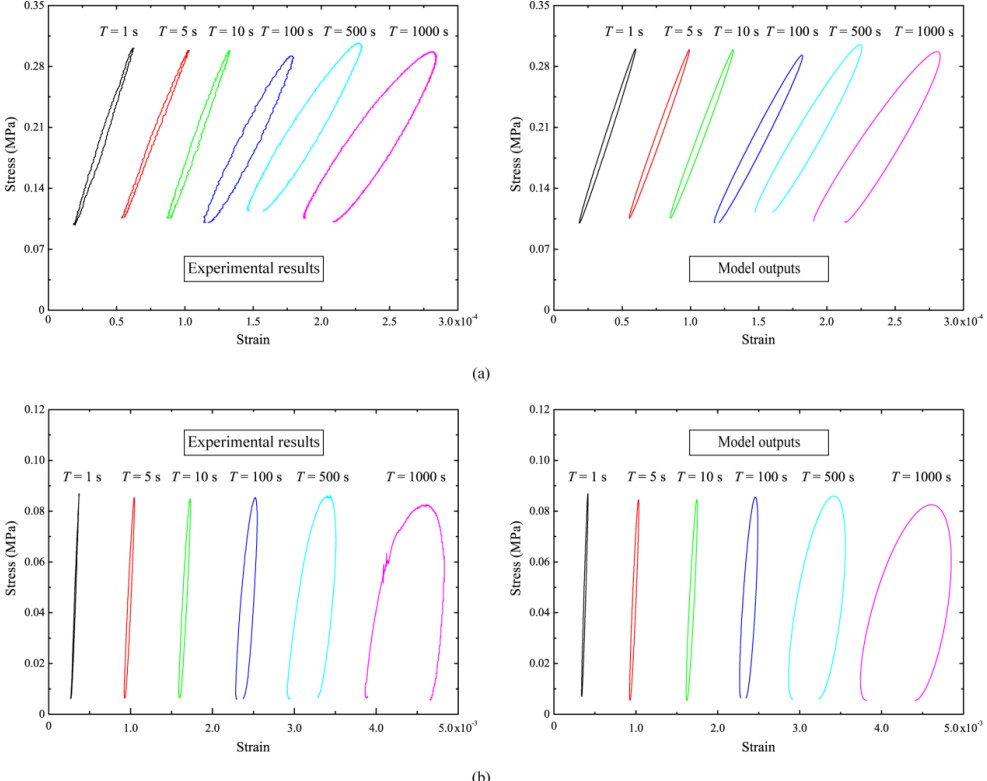

(a)

(b)

**Figure 14.** Comparison between the steady-state stress-strain hysteresis loops measured in the experiments and those predicted by the dislocation-based model for (a) specimen Dry-5ppt-1 tested with stresses varying from 0.1 to 0.3 MPa and with the temperature of -10°C and (b) specimen Wet-5ppt-1 tested with stresses varying from 0.005 to 0.085 MPa and with the average temperature of -2.5°C (note that, to compare the representative hysteresis loops in all the different-frequency experiments with those outputted by the model in a more concise and intuitive way, the pre-strains before the hysteresis loops drawn here are not equal to the experimental values).
