# Peer review of "Strain response and energy dissipation of floating saline ice"

_The Cryosphere, 2020_

## Referee Comment (RC1) · Anonymous Referee #1 · 13 Mar 2020

This is an interesting paper about structured and well-described experiments on the cyclic loading of saline ice. In this work, novel and apparent test setup is used for testing of floating ice in the laboratory. Saline ice is produced in the laboratory in an unusual way and, surprisingly, showed a microstructure very similar to that of S2 sea ice. The growing and preparation method ensures that ice structure and the presence of brine in ice are not affected until the experiment begins. A fair amount of discussion is presented concerning the earlier work, suggesting reasons for the need in the testing of floating wet ice samples. The behavior of ice upon cycling is well predicted by the model. The key point of the paper is that warm wet floating ice behaves differently from cold dry ice. A weakness of the paper is that there is no clear answer/evidence on whether both water and temperature or only temperature play a major role in the

mechanical behavior of ice under cyclic loading.

Overall, the paper is clearly written and provides new results. It is worthy of publication once some details have been clarified.

Specific comments:

1. Please, state that the sinusoidal waveform was used during cycling in the abstract (line 13) and introduction (line 61) for the readers' convenience. For example, "stress-controlled sinusoidal cyclic compression experiments" (lines 13-14).

2. Lines 38-39: A reference to the study of ice fatigue in-situ tests led by Langhorne shall be provided. For example:

* Bond PE and Langhorne PJ (1997) Fatigue behavior of cantilever beams of saline ice. J. Cold Reg. Eng. 11(2), 99–112;

* Haskell TG, Robinson WH and Langhorne PJ (1996) Preliminary results from fatigue tests on in situ sea ice beams. Cold Reg. Sci. Technol. 24(2), 167–176

* Langhorne PJ, Squire VA, Fox C and Haskell TG (1998) Break-up of sea ice by ocean waves. Ann. Glaciol. 27, 438–442

3. Lines 40-42: Strictly speaking, first cyclic loading experiments on freshwater ice were conducted in the forties: Kartashkin B.D., 1947. Experimental studies of the physico-mechanical properties of ice. Similarly, experiments on sea ice were firstly performed in the eighties: Tabata T, Nohguchi Y, 1980. Failure of sea ice by repeated compression.

4. Lines 54-57: Reference to in-situ experiments by Langhorne shall be provided.

5. Lines 92-93: What was the reason for the temperature to be changed twice?

6. Line 99: What does "about" mean? Can authors provide standard deviation or standard error for their measurements?
7. Line 127: What was the accuracy of temperature measurements? Thermistors and thermocouples usually have an accuracy in the range from about ±0.3 to ±2.5°C. In this case, the resolution is not important for the manuscript and shall be replaced with accuracy.

8. Line 137: Again, what is the accuracy of LVDT? Is it more important than a resolution?

9. The stress during cycling was as low as 0.005-0.085 MPa (line 327). This range seems to be very low. What was the accuracy of a load cell and how accurate the machine (actuator) could control load-limits? Could the test setup ensure accurate cycling between 0.005 and 0.085 MPa? This should be commented on in the text.

10. When using words "linear loading" you always should be careful since ice never behaves purely elastically (linearly); an inelastic component (though minor) is always present.

11. Lines 212-213: References to other works that show similarly that the hysteresis loop area increases with an increase of the cyclic period shall be added:

* Weber LJ and Nixon WA (1996) Hysteretic Behavior in Ice Under Fatigue Loading. Proceedings of the 15th International Conference on Offshore Mechanics and Arctic Engineering. 75–82

* Murdza A, Schulson EM and Renshaw CE (2018) Hysteretic behavior of freshwater ice under cyclic loading : preliminary results. 24th IAHR International Symposium on Ice. Vladivostok, 185–192

* Cole DM (1990) Reversed direct-stress testing of ice: Initial experimental results and analysis. Cold Reg. Sci. Technol. 18(3), 303–321.

12. It is mentioned in Lines 208-211 that the area of the hysteresis loop is decreasing until a "steady-state" is reached. Does this happen only during the first set of loading (T=1s) or during any subsequent loadings as well? (especially after 15 min of recovery

in the case of dry ice)? If some cycles are needed to reach a steady-state condition every time (for example after relaxation) then is N=4 cycles at T=1000s for dry ice and N=1 cycle for wet ice enough to get a steady-state as mentioned in line 225?

13. It is emphasized through the manuscript on the importance of considering warm floating ice for the experiments, in contrast to cold dry ice. In addition, the conclusion that water and temperature have a greater effect on elastic modulus than salinity is made. It is not a big surprise that "warm" ice behaves differently than "cold" ice and the temperature of ice affects elastic modulus. Therefore, do you think that if you repeat your experiments on dry ice at about -2.5°C (average temperature of wet specimens based on line 153) instead of -10°C it would behave similarly to wet specimens? In this case, no additional brine will freeze during the storage as mentioned in line 410. Is it possible that brine migration during cycling affects mechanical properties? Perhaps, there is no need to conduct experiments on floating ice but rather increase the ice temperature. If authors think similarly, they should state it more clear because the reader may get the impression that both floating and warm conditions are equally important during cyclic loading of ice (which may not be true). I think it would be interesting to compare the results of both wet and dry ice of similar temperatures.

14. Generally, a paper should be short and laconic but "full" in context. I suggest the authors make their manuscript shorter where it is possible by removing unnecessary parts. For example, in lines 100-101: "The specimens used in the dry experiments (Figure 2a and 3) were sealed in plastic bags and stored in a freezer for 1-2 days before testing. The freezer temperature was set to -10℃." can be replaced as: "The specimens used in the dry experiments (Figure 2a and 3) were sealed in plastic bags and stored in a freezer at -10°C for 1-2 days before testing.".

15. Figure 4: Ice salinity shall be mentioned in the caption.

16. Line 89: The verb "nucleated" fits better than "generated".

---

## Referee Comment (RC2) · Anonymous Referee #2 · 9 Apr 2020

650

[referee-annotated manuscript omitted]

---

## Referee Comment (RC3) · Anonymous Referee #2 · 21 Apr 2020

**Strain response and energy dissipation of floating saline ice**
**under cyclic compressive stress**
by Mingdong Wei, Arttu Polojärvi, David M. Cole, and Malith Prasanna

*Submitted to The Cryosphere Discussions*

*Review*
*April 9, 2020*

**Summary**

This manuscript presents results from lab measurements and numerical models of the strain experience saline ice undergoing periodic compressive stress. The experimental set up is novel in that allows the ice sample to be immersed in water while stress is exerted by a electrohydraulic cylinder. This allows a vertical temperature profile to be maintained through the sample, which is more representative of sea ice conditions found outside the laboratory. The authors find measurable differences in the cumulative strain response between "wet" and "dry" experiments for all frequencies of periodic loading, which particularly significant differences at low frequencies. At a loading period of 1000s, the "dry" ice samples showed only 24% of the energy density dissipation as the "wet" samples. Using a dislocation-based model initially developed by co-author Cole and others, the authors are able to qualitatively and quantitatively reproduce the experimental results by assuming a significantly lower elastic modulus ($E_0$), a higher dislocation density ($\rho$), and a higher dislocation relaxation strength ($\delta D^d$) for "wet" ice than "dry" ice.

Overall, the manuscript is well written the figures are clear and well labeled. The measured difference in strain response between "dry" and "wet" experiments suggests the non-isothermal temperature profile of immersed ice under a cold atmosphere has a significant effect on mechanical behavior, which should be considered in future experiments. However, I feel that some additional explanation is required regarding the methods used to determine the values of $E_0$, $\rho$ and $\delta D^d$. I believe the manuscript would also benefit from a deeper discussion of the temperature dependence of these and other parameters used in the model. My only other significant comment concerns the usage of the term "floating" in the manuscript. These comments are described in detail below, together with a list of minor comments for specific lines of text. In sum, I believe these amount to more than just minor revisions, but I feel they should all be quite straightforward to address.

**Major Comments**

1. Unclear derivation of model parameters
   The should provide the reader with more information about the empirical method used to determine the values for $E_0$, $\rho$ and $\delta D^d$ listed in Table 4. Line 300 mentions a "trial and error" method, but it is not clear if applies just to $E_0$ or other model parameters as well. Also, the text states on lines 295-296 that values for $\delta D^{gb}$ were determined empirically, but these are not listed in Table 4 or mentioned elsewhere in the text. The authors should describe in detail the method used to determine the values for each parameter and provide an assessment of the sensitivity of the model to each parameter.

2. Greater discussion of temperature dependence of mechanical behavior of saline ice
   The difference in the observed strain response between wet and dry samples is attributed to the higher temperature of the wet ice, while the model results indicate that the difference is due to a lower elastic modulus, $E_0$, and higher dislocation density, $\rho$. However, the connection between these

parameters and the temperature of the ice is not made clear. I recommend the authors expand their discussion to give the reader further insight into the temperature dependence of these two parameters. Also, the viscous strain rate, $\dot{\varepsilon}_v$, is the only parameter specifically identified as having a temperature dependence (equation 10) and so I was surprised not to see greater discussion of this in the text.

3. Use of the phrase "floating ice"
   I have two minor concerns with the use of the term "floating ice" in the manuscript:
   a) First, I wonder whether the ice sample in the wet experiments can really be considered to be floating once the compressive stress is applied. If the water level were changed during the experiment, the sample would presumably not rise or fall. So, I wonder whether "immersed" would be a more appropriate term to use.
   b) Second, in the discussion and conclusions section the phrase "floating ice" sometimes appears to be used to refer more generally to real world ice outside the laboratory. Specifically, on line 373, the phrase is used almost synonymously with "full-scale". I recommend that the authors add additional language to clarify that the "wet" lab experiments are able to replicate the temperature profile of floating ice, but not necessarily all the other ways in which the real world differs from the lab. See also specific comments below referring to lines 298 and 350.

**Specific comments**

Lines 51-52: This statement is not strictly accurate. One the air temperature rises above freezing in spring, the ice will approach an isothermal state

Line 83: It is not necessary or accurate to refer to the "bulk" salinity of seawater. The word "bulk" can be deleted.

Line 106: Sea ice literature more commonly describes this microstructure as "non-oriented columnar". For readers not familiar with the designators S2, S3, etc, I recommend the authors add some brief text explaining the relevant microstructures.

Lines 206-207: This feature of the data could be highlighted with additional annotation. Also, the total amount of strain also seems to increase with loading period, with the exception of T=10s, which seems to yield less strain than T=1s or T=5s. Can the authors comment on this?

Line 221: Misplaced comma after "both"

Line 283: there should be a citation here for the value of $\Omega$ for unaligned S2 saline ice.

Line 293: For clarity, I recommend adding "($\rho$)" after "dislocation density"

Line 298: Are the authors referring to field or lab measurements of floating ice here? Please also refer to general comment 1b above

Line 316: Should "modes" be "models"?

Line 350: A citation would be appropriate here. Additionally, it would be helpful to clarify whether the authors are refering to lab- or field-scale observations of the elastic modulus of floating ice (see also comment 1b above).

Lines 381-383: By using the phrase "the water and the related through-thickness temperature gradient", the authors appear to suggest that water itself (and not just the resulting change in temperature profile) exerts some influence on the elastic modulus of the ice. Further clarification of this statement is needed.

---

## Author Comment (AC1) · 21 May 2020

Summary

This manuscript presents results from lab measurements and numerical models of the strain experience saline ice undergoing periodic compressive stress. The experimental set up is novel in that allows the ice sample to be immersed in water while stress is exerted by a electrohydraulic cylinder. This allows a vertical temperature profile to be maintained through the sample, which is more representative of sea ice conditions found outside the laboratory. The authors find measurable differences in the cumulative strain response between "wet" and "dry" experiments for all frequencies of periodic loading, which particularly significant differences at low frequencies. At a loading pe-

riod of 1000s, the "dry" ice samples showed only 24% of the energy density dissipation as the "wet" samples. Using a dislocationbased model initially developed by co-author Cole and others, the authors are able to qualitatively and quantitatively reproduce the experimental results by assuming a significantly lower elastic modulus (E0), a higher dislocation density (ðİlJŇ), and a higher dislocation relaxation strength (ðİŽ£ðİŘůd) for "wet" ice than "dry" ice.

Overall, the manuscript is well written the figures are clear and well labeled. The measured difference in strain response between "dry" and "wet" experiments suggests the non-isothermal temperature profile of immersed ice under a cold atmosphere has a significant effect on mechanical behavior, which should be considered in future experiments. However, I feel that some additional explanation is required regarding the methods used to determine the values of E0, ðİlJŇ and ðİŽ£ðİŘůd. I believe the manuscript would also benefit from a deeper discussion of the temperature dependence of these and other parameters used in the model. My only other significant comment concerns the usage of the term "floating" in the manuscript. These comments are described in detail below, together with a list of minor comments for specific lines of text. In sum, I believe these amount to more than just minor revisions, but I feel they should all be quite straightforward to address.

Re: We sincerely thank the reviewer for the encouragement on our work. The comments are constructive and insightful. We have modified our manuscript according to them.

Major Comments

1. Unclear derivation of model parameters

The should provide the reader with more information about the empirical method used to determine the values for E0, ðİlJŇ and ðİŽ£ðİŘůd listed in Table 4. Line 300 mentions a "trial and error" method, but it is not clear if applies just to E0 or other model parameters as well. Also, the text states on lines 295-296 that values for ðİŽ£ðİŘůgb

were determined empirically, but these are not listed in Table 4 or mentioned elsewhere in the text. The authors should describe in detail the method used to determine the values for each parameter and provide an assessment of the sensitivity of the model to each parameter.

Re: We thank the reviewer for the constructive comment. More information (as listed below) about the empirical method used to determine the values for E0, dislocation density and grain boundary relaxation strength, and an assessment of the sensitivity of the model to each parameter, has been added in the revised manuscript (as listed below). Please note that dislocation relaxation strength is dependent on dislocation density (Eq. (8)); once dislocation density is determined, dislocation relaxation strength is also known. Values for grain boundary relaxation strength could be found in Table 4.

Lines 271-279, "By making the slopes of the modeled and experimental hysteresis loops for T = 10 s comparable, E0 could be determined. This is because the behavior of the specimens is mainly dominated by the un-relaxed modulus E0 when the loading frequency is high (here 0.1 Hz to 1 Hz), as indicated in Figures 13 and 14. From Eqs. (9) and (10), one can find that the strain increment under one loading cycle is dependent on the dislocation density; based on this, the dislocation density was estimated by using the experimental results of T = 100 and 500 s and the dislocation relaxation strength was then determined from Eq. (8). The grain boundary relaxation strength was determined by referring to previous work (Cole 1995) because it could be reasonably assumed constant for the ice material of interest here, and its effect on inelastic behavior of ice was significantly less than the dislocation mechanism"

Lines 383-386, "It was also found that for relatively high loading frequency (for example, 1 Hz used here), the modeled strain behavior was dominated by the un-relaxed modulus E0, not sensitive to the dislocation density or the strength of grain boundary relaxation. However, for low-frequency (0.001 Hz) cyclic loading, the modeled specimen deformation was very sensitive to the dislocation density."
**TCD**

2. Greater discussion of temperature dependence of mechanical behavior of saline ice

The difference in the observed strain response between wet and dry samples is attributed to the higher temperature of the wet ice, while the model results indicate that the difference is due to a lower elastic modulus, E0, and higher dislocation density, ðİIJŇ. However, the connection between these parameters and the temperature of the ice is not made clear. I recommend the authors expand their discussion to give the reader further insight into the temperature dependence of these two parameters. Also, the viscous strain rate, , is the only parameter specifically identified as having a temperature dependence (equation 10) and so I was surprised not to see greater discussion of this in the text.

Re: We thank the reviewer for the insightful comment. We have expanded our discussion to give readers further insight into the temperature dependence of E0 and dislocation density. Details can be found in the revised manuscript (as shown below).

Lines 392-396, "The analysis and modeling indicated that the physical mechanisms of deformation in both the warmer, floating specimens and the colder dry specimens were essentially the same. Warmer saline ice had a smaller modulus due to its higher liquid brine volume, which necessarily decreases the volume of the solid ice matrix (thereby reducing the bulk elastic modulus) and there is a pronounced increase in the effective dislocation density with increasing temperature (Cole and Durell, 2001; Timco and Weeks, 2010; Cole, 2020)."

3. Use of the phrase "floating ice"

I have two minor concerns with the use of the term "floating ice" in the manuscript:

a) First, I wonder whether the ice sample in the wet experiments can really be considered to be floating once the compressive stress is applied. If the water level were changed during the experiment, the sample would presumably not rise or fall. So, I wonder whether "immersed" would be a more appropriate term to use.

Re: Thanks for this comment. Before the ice specimen is compressed, it floats naturally on water. From the strain response of the ice, the change of water level can be determined to be in the magnitude of only 0.001 mm. Thus, it can be approximated that the specimen is still floating.

b) Second, in the discussion and conclusions section the phrase "floating ice" sometimes appears to be used to refer more generally to real world ice outside the laboratory. Specifically, on line 373, the phrase is used almost synonymously with "full-scale". I recommend that the authors add additional language to clarify that the "wet" lab experiments are able to replicate the temperature profile of floating ice, but not necessarily all the other ways in which the real world differs from the lab. See also specific comments below referring to lines 298 and 350.

Re: Thanks for the recommendation. Corresponding changes have been made in the revised manuscript (as shown below). More responses can also be found below the related specific comments.

Lines 355-358, "Even if the use of floating specimens could be considered to only address the temperature profiles of in-situ floating ice (with some other environmental conditions of natural ice floes ignored)..."

Specific comments

Lines 51-52: This statement is not strictly accurate. One the air temperature rises above freezing in spring, the ice will approach an isothermal state

Re: Thanks for pointing this out. We have modified the statement in the revised manuscript.

Line 51, "Floating ice commonly has a through-thickness temperature gradient"

Line 83: It is not necessary or accurate to refer to the "bulk" salinity of seawater. The word "bulk" can be deleted.

Re: The corresponding change has been made in the revised manuscript. Thanks.

Line 106: Sea ice literature more commonly describes this microstructure as "non-oriented columnar". For readers not familiar with the designators S2, S3, etc, I recommend the authors add some brief text explaining the relevant microstructures.

Re: Thanks for pointing this out. We have improved the text to avoid using the designators S1, S2 and S3.

For example, Line 61, "non-oriented columnar saline ice specimens"

Lines 206-207: This feature of the data could be highlighted with additional annotation. Also, the total amount of strain also seems to increase with loading period, with the exception of T=10s, which seems to yield less strain than T=1s or T=5s. Can the authors comment on this?

Re: Information on this is added to the revised manuscript.

Lines 177-180, "The total amount of strain does not strictly increase with the loading period. This may be because the loading platen and the specimen did not always achieve perfect contact immediately, causing some error in the strain measured in this initial stage of loading. Once intimate contact was achieved, the measured strain became reliable."

Line 221: Misplaced comma after "both"

Re: The sentence has been modified. Thanks.

Line 194, "both the strain increment per cycle and the area of one hysteresis loop"

Line 283: there should be a citation here for the value of $\Omega$ for unaligned S2 saline ice.

Re: Thanks for pointing this out. The corresponding change has been made in the revised manuscript. Line 256, "$\Omega = 1/\pi \approx 0.32$ for a horizontal specimen made of unaligned columnar ice (Cole, 1995)"

[Figure]

Line 293: For clarity, I recommend adding "(Rho)" after "dislocation density"

Re: The corresponding change has been made in the revised manuscript. Thanks.

Line 267, "the dislocation density (Rho)"

Line 298: Are the authors referring to field or lab measurements of floating ice here? Please also refer to general comment 1b above

Re: Originally, the statement referred to field measurements of floating ice. Since more detailed information on how the parameter values were determined has now been provided, some redundant descriptions, including the sentence discussed in this comment, have been deleted from the revised manuscript.

Line 316: Should "modes" be "models"?

Re: Apologies for the typo. The error has been corrected in the revised manuscript.

Line 350: A citation would be appropriate here. Additionally, it would be helpful to clarify whether the authors are refering to lab- or field-scale observations of the elastic modulus of floating ice (see also comment 1b above).

Re: Corresponding changes have been made in the revised manuscript.

Line 332, "...as would be expected based on full-scale observations on the effect of temperature on this property (Timco and Weeks 2010..."

Lines 381-383: By using the phrase "the water and the related through-thickness temperature gradient", the authors appear to suggest that water itself (and not just the resulting change in temperature profile) exerts some influence on the elastic modulus of the ice. Further clarification of this statement is needed.

Re: Thanks for pointing this out. We have modified the manuscript to make its main idea clearer by removing unnecessary and potentially misleading descriptions. The phrase mentioned here is deleted from the manuscript.

---

## Author Comment (AC2) · 21 May 2020

This is an interesting paper about structured and well-described experiments on the cyclic loading of saline ice. In this work, novel and apparent test setup is used for testing of floating ice in the laboratory. Saline ice is produced in the laboratory in an unusual way and, surprisingly, showed a microstructure very similar to that of S2 sea ice. The growing and preparation method ensures that ice structure and the presence of brine in ice are not affected until the experiment begins. A fair amount of discussion is presented concerning the earlier work, suggesting reasons for the need in the testing of floating wet ice samples. The behavior of ice upon cycling is well predicted by the model. The key point of the paper is that warm wet floating ice behaves differently from cold dry ice. A weakness of the paper is that there is no clear answer/evidence

on whether both water and temperature or only temperature play a major role in the mechanical behavior of ice under cyclic loading.

Overall, the paper is clearly written and provides new results. It is worthy of publication once some details have been clarified.

Re: We sincerely thank the reviewer for the constructive comments and valuable time devoted to improving our manuscript. We have modified our manuscript according to the comments. The majority of them led to modifications.

Specific comments:

1. Please, state that the sinusoidal waveform was used during cycling in the abstract (line 13) and introduction (line 61) for the readers' convenience. For example, "stress-controlled sinusoidal cyclic compression experiments" (lines 13-14).

Re: Thanks for the comment. We made the corresponding changes in the revised manuscript.

Lines 13 and 14, "stress-controlled sinusoidal cyclic compression experiments"

2. Lines 38-39: A reference to the study of ice fatigue in-situ tests led by Langhorne shall be provided. For example: * Bond PE and Langhorne PJ (1997) Fatigue behavior of cantilever beams of saline ice. J. Cold Reg. Eng. 11(2), 99–112; * Haskell TG, Robinson WH and Langhorne PJ (1996) Preliminary results from fatigue tests on in situ sea ice beams. Cold Reg. Sci. Technol. 24(2), 167–176 * Langhorne PJ, Squire VA, Fox C and Haskell TG (1998) Break-up of sea ice by ocean waves. Ann. Glaciol. 27, 438–442.

Re: We thank the reviewer for pointing this out. Corresponding changes are made in the revised manuscript.

Line 37, "...give insight into the fatigue of ice (Bond and Langhorne, 1997; Langhorne et al.,1998...)"

3. Lines 40-42: Strictly speaking, first cyclic loading experiments on freshwater ice were conducted in the forties: Kartashkin B.D., 1947. Experimental studies of the physico-mechanical properties of ice. Similarly, experiments on sea ice were firstly performed in the eighties: Tabata T, Nohguchi Y, 1980. Failure of sea ice by repeated compression.

Re: The literature review is modified in the revised manuscript.

Lines 40 and 41, "...have been performed since the forties (Kartashkin, 1947; Mellor and Cole, 1981) and on saline ice since the eighties (Tabata and Nohguchi, 1980..."

4. Lines 54-57: Reference to in-situ experiments by Langhorne shall be provided.

Re: Three pieces of literature by Langhorne, related to in-situ experiments, are added in the revised manuscript.

Line 54, "...in-situ experiments on floating ice (Langhorne et al., 2015; Smith et al., 2015; Wongpan et al., 2018)"

5. Lines 92-93: What was the reason for the temperature to be changed twice?

Re: Lower temperature (-14 degrees Celsius) was used for practical reason: make the ice grow faster. This was then changed to -10 degrees Celsius to perform the tests in a temperature such tests are often performed. The change was done well ahead of the actual experiments to ensure that the ice used for floating experiments had stable temperature and thermal gradient. We did not notice differences in the ice structure due to the change.

6. Line 99: What does "about" mean? Can authors provide standard deviation or standard error for their measurements?

Re: The standard deviations for the density measurements are provided in the revised manuscript.

Line 88, "...and their densities were 886±19 and 879±16 kgÂům-3, respectively"

7. Line 127: What was the accuracy of temperature measurements? Thermistors and thermocouples usually have an accuracy in the range from about ±0.3 to ±2.5åŮęC. In this case, the resolution is not important for the manuscript and shall be replaced with accuracy.

Re: We agree. The accuracy of temperature measurements is ±0.5 degrees Celsius. This information is supplemented in the revised manuscript. Thanks for this constructive comment.

Line 112, "...and an accuracy of ± 0.5 degrees Celsius"

8. Line 137: Again, what is the accuracy of LVDT? Is it more important than a resolution?

Re: The description is updated in the revised manuscript (line 144). We agree that the accuracy of LVDT (0.001 mm) is more important than the resolution (0.0001 mm). The "resolution" described in the original manuscript actually refers to the "accuracy". Apologies for this clerical error.

Line 120, "...with a measurement range and accuracy of 2 and ± 0.001 mm, respectively"

9. The stress during cycling was as low as 0.005-0.085 MPa (line 327). This range seems to be very low. What was the accuracy of a load cell and how accurate the machine (actuator) could control load-limits? Could the test setup ensure accurate cycling between 0.005 and 0.085 MPa? This should be commented on in the text.

Re: We modified the text to comment on this. Since our specimen size was 0.6 m × 0.3 m × 0.1 m, the force applied for stress amplitude 0.005–0.085 MPa was 0.15–2.55 kN. The accuracy of the load cell was ±5 N; thus, there was no severe error on the cyclic stress values and we have a reason to assume the cycling was accurate.

Line 106, "The load cell had an accuracy of ± 5 N, which is sufficient for all stress levels and cycles of the experiments here"

Line 305, "Nominal cyclic stress of 0.005–0.085 MPa is low, but the setup could achieve it: With the accuracy of the system, the actual stress applied to the specimen was 0.005 ($\pm$0.001)–0.085 ($\pm$0.003) MPa"

10. When using words "linear loading" you always should be careful since ice never behaves purely elastically (linearly); an inelastic component (though minor) is always present.

Re: We agree. The statement was actually referring to the application of a linear loading ramp, not the response of the ice. For clarity, the related sentences have been modified as follows.

Line 144, "the duration of the initial loading ramp was fixed to be 1 s"

11. Lines 212-213: References to other works that show similarly that the hysteresis loop area increases with an increase of the cyclic period shall be added: * Weber LJ and Nixon WA (1996) Hysteretic Behavior in Ice Under Fatigue Loading. Proceedings of the 15th International Conference on Offshore Mechanics and Arctic Engineering. 75–82 * Murdza A, Schulson EM and Renshaw CE (2018) Hysteretic behavior of fresh-water ice under cyclic loadingâÄŸ ′r: preliminary results. 24th IAHR International Symposium on Ice. Vladivostok, 185–192 * Cole DM (1990) Reversed direct-stress testing of ice: Initial experimental results and analysis. Cold Reg. Sci. Technol. 18(3), 303–321.

Re: Missing references have been added in the revised manuscript.

Line 187, "…consistent with earlier studies (Cole, 1990; Murdza et al., 2018; Weber and Nixon, 1996)"

12. It is mentioned in Lines 208-211 that the area of the hysteresis loop is decreasing until a "steady-state" is reached. Does this happen only during the first set of loading (T=1s) or during any subsequent loadings as well? (especially after 15 min of recovery in the case of dry ice)? If some cycles are needed to reach a steady-state condition

every time (for example after relaxation) then is N=4 cycles at T=1000s for dry ice and N=1 cycle for wet ice enough to get a steady-state as mentioned in line 225?

Re: In brief, the phenomenon happened in all the dry experiments. However, the larger the T value, the less the number of cycles required to reach the "steady-state". This is shown by Figure 9, in which there are 5 or 6 cycles before the experiment with T=100 s reaches the "steady-state", while the experiments with T = 500 and 1000 s only require two cycles and one cycle, respectively, to achieve the "steady-state". An important reason why the dry specimens require some cycles to reach the "steady-state" is the 15-minute recovery period before each experiment. In the floating ice experiments, the cyclic loads with increasing periods were applied to the specimen in a continuous manner – loading with one period after another without a recovery period. After some initial loading cycles of T = 1 s, the ice samples maintained a relatively steady state in subsequent cyclic loads (Figures 10 and 11). For example, the areas of the two hysteresis loops at T = 100 s are very similar, having a much smaller difference than those of the first two hysteresis loops in the dry experiments with T = 100 s. Therefore, it is believed that N = 1 can be used for the floating experiment with T = 1000 s. In addition, the good agreement between the experimental and modeling results also shows the reliability of the experimental scheme and test results to a certain extent. We modified the text to discuss this, as shown below.

Line 183, "For example, the hysteresis loops after the first stress cycle in the dry experiment with T = 1000 s are similar; N = 4 is enough for the dry specimen at T = 1000 s to show steady-state response"

13. It is emphasized through the manuscript on the importance of considering warm floating ice for the experiments, in contrast to cold dry ice. In addition, the conclusion that water and temperature have a greater effect on elastic modulus than salinity is made. It is not a big surprise that "warm" ice behaves differently than "cold" ice and the temperature of ice affects elastic modulus. Therefore, do you think that if you repeat your experiments on dry ice at about -2.5åŮęC (average temperature of wet specimens

based on line 153) instead of -10åŮ ęC it would behave similarly to wet specimens? In this case, no additional brine will freeze during the storage as mentioned in line 410. Is it possible that brine migration during cycling affects mechanical properties? Perhaps, there is no need to conduct experiments on floating ice but rather increase the ice temperature. If authors think similarly, they should state it more clear because the reader may get the impression that both floating and warm conditions are equally important during cyclic loading of ice (which may not be true). I think it would be interesting to compare the results of both wet and dry ice of similar temperatures.

Re: Earlier laboratory work on ice has mostly focused on dry, cold and isothermal ice. Here we wish to draw attention to the fact, that the results from such experiments may not apply directly on saline ice in its natural conditions, that is, when ice is floating in water – and also to the fact, that the experiments where this type of conditions are mimicked in the laboratory can be performed. We do not aim to differentiate between the importance of the different factors on the ice behavior and hope this is now more clearly stated in the manuscript. In brief, our goal was (i) to present our methods used to perform laboratory-scale experiments on ice in its natural conditions and validate their applicability with floating ice experiments and commonly used test conditions (dry experiments), (ii) to report experimental results from cyclic loading tests on floating and dry specimens using different salinities, stress amplitude and loading-unloading periods, and (iii) to analyze the test results of the floating ice and dry ice specimens using a physically based model.

Thus, the original intention of this study is not to investigate the effect of temperature and floating condition on ice behavior. Probably the previous version of our manuscript made some excessive comparisons between warm floating ice and cold dry ice and some sentences were somewhat misleading in that regard. The revised manuscript clarifies our goals. For example, we reorganized the conclusions around the two themes of this article: experimental methods and material modeling of floating ice. We also avoided using "wet experiments" to refer to the floating ice experiments, in

order to avoid the misunderstanding that here we are studying the effect of water on ice behavior. We thank the reviewer this constructive comment.

Comparing the results of both wet and dry "warm" ice of would be interesting, but would need another extensive experimental campaign. Inspired by the comments by the reviewer, we plan to conduct experiments on this in our future work. In fact, research on the effect of water is scarce, which also shows the necessity of developing experimental equipment for floating ice tests (as done here), which will help to reveal the mechanical properties of floating ice more deeply in the future.

14. Generally, a paper should be short and laconic but "full" in context. I suggest the authors make their manuscript shorter where it is possible by removing unnecessary parts. For example, in lines 100-101: "The specimens used in the dry experiments (Figure 2a and 3) were sealed in plastic bags and stored in a freezer for 1-2 days before testing. The freezer temperature was set to -10âDËĞ C." can be replaced as: "The ËĞ specimens used in the dry experiments (Figure 2a and 3) were sealed in plastic bags and stored in a freezer at -10◦C for 1-2 days before testing.".

Re: Thanks for this constructive comment. We have modified the manuscript by removing some unnecessary parts.

15. Figure 4: Ice salinity shall be mentioned in the caption.

Re: Ice salinity is now provided in the caption of Figure 4.

Caption of Figure 4, ". . .thin sections of the ice (salinity: 5 ppt) . . ."

16. Line 89: The verb "nucleated" fits better than "generated".

Re: Thanks for pointing this out. Since this part of description is unnecessary, according to your Comment 14, we deleted it.

---

## Author Comment (AC3) · 21 May 2020

Because Anonymous Referee #2 states that a wrong PDF document was attached here, please see authors' responses to the updated comments from the referee. Thanks!

---

## Referee Report (RR1)

**Reviewer's Comments on the MS: "Strain response and energy dissipation of floating saline ice under cyclic compressive stress"**

The manuscript was well revised according to the reviewers' comments. The new version of the manuscript clearly provides new methods along with the obtained results and does not confuse the reader whether temperature or other "immersed" conditions play a more important role in the ice behavior under cycling. The new Conclusion is divided into two parts and differentiate well between the novel test setup and main results that were obtained.

The results on cyclic loading of floating ice specimens obtained through the novel well-designed experimental methods & setup must be important for the further development of research on the mechanical properties of ice. The underlying physical processes are described and give a good agreement with numerical simulations.

Therefore, I recommend publication of this manuscript in "The Cryosphere".

---

## Referee Report (RR2)

**Strain response and energy dissipation of floating saline ice
under cyclic compressive stress**
by Mingdong Wei, Arttu Polojärvi, David M. Cole, and Malith Prasanna

*Submitted to The Cryosphere Discussions*

**2nd Review**
*July 7, 2020*

**Summary**
The authors have responded to the majority of my comments from my first review and I believe the manuscript has been improved as a result. In particular, I am happy to see more details provided on the method used to estimate values for $E_0$ and $\rho$ and to see a clearer explanation of the temperature dependence of these variables and their relationship to the observed higher rates of strain for warmer "floating" ice. However, I am still surprised to find no discussion of the temperature dependence of viscous strain. I am also puzzled by the authors' response to one of reviewer 1's comments that the "original intention of this study is not to investigate the effect of temperature and floating condition on ice behavior". I feel this understates the authors' work and the resulting changes to the text detract from the overall quality of the manuscript.

**Major comments**

**1. Still no discussion of temperature dependence of viscous strain component**
According to equation 10, the differences in ice temperature between the wet and dry experiments ought to result in an approximately two-fold difference in the viscous component of strain, $\varepsilon_v$. This should be discussed on more detail in relation to the overall increase in strain observed in the "floating" ice.

**2. Ambiguous premise of paper**
In their response to reviewer 1's comment 13, concerning the processes being tested by the experiment design, the authors state that "*the original intention of this study is not to investigate the effect of temperature and floating condition on ice behavior*" and the manuscript should instead be considered as a methods paper, as reflected by a minor restructuring of the conclusions section. I find this to be a disappointing climb-down, when I would have preferred to see a clarification of the conclusions that can be drawn from the authors' novel laboratory method.

I also find this to be an entirely unconvincing argument. If the study was intended to be an investigation into the effects of temperature an floating conditions, why did the authors include experiments on both cold, isothermal non-floating ice and poly-thermal floating ice? Instead of removing references "wet" experiments, I urge the authors to revert to their nomenclature and instead spend a little extra text speculating on how floating, polythermal ice might respond differently to non-floating ice with an equivalent average temperature.

In its current form, the premise of the manuscript is ambiguous and despite removing references to "wet" experiments, the text still contains phrases like "Thus, the fact that the ice was in contact with water and had a realistic temperature profile …" (line 365), which still contribute to the misunderstanding that the authors are studying the effect of water on ice behavior.

---

## Author Response (AR2)

**Authors' Response**

**Editor Decision**

Comments to the Author:

Dear Mingdong Wei, based on positive feedback from the two referees, I am pleased to formally accept your paper for its publication with minor revisions in The Cryosphere.

Owing to constructive remark of the Referee #2 (about the way the effect of temperature is discussed in the manuscript), I am choosing "publish subject to minor revisions" and encourage you to consider this remark very carefully as your prepare your final text, which I will go through one more time.

Thank you for your patience with this slow process, but all of us are dealing with multiple issues these days.

Best regards,

Evgeny Podolskiy

Re: We sincerely thank Prof. Podolskiy very much for the valuable time devoted to improving our manuscript. We have further modified the manuscript according to the comments from Referee #2.

**Referee #1**

The manuscript was well revised according to the reviewers' comments. The new version of the manuscript clearly provides new methods along with the obtained results and does not confuse the reader whether temperature or other "immersed" conditions play a more important role in the ice behavior under cycling. The new Conclusion is divided into two parts and differentiate well between the novel test setup and main results that were obtained.

The results on cyclic loading of floating ice specimens obtained through the novel well-designed experimental methods & setup must be important for the further development of research on the mechanical properties of ice. The underlying physical processes are described and give a good agreement with numerical simulations.

Therefore, I recommend publication of this manuscript in "The Cryosphere".

Re: We thank the reviewer for the insightful comments and valuable time.

**Referee #2**

Summary

The authors have responded to the majority of my comments from my first review and I believe the manuscript has been improved as a result. In particular, I am happy to see more details provided on the method used to estimate values for E0 and r and to see a clearer explanation of the temperature dependence of these variables and their relationship to the observed higher rates of strain for warmer "floating" ice. However, I am still surprised to find no discussion of the temperature dependence of viscous strain. I am also puzzled by the authors' response to one of reviewer 1's comments that the "original intention of this study is not to investigate the effect of temperature and floating condition on ice behavior". I feel this understates the authors' work and the resulting changes to the text detract from the overall quality of the manuscript.

Re: We thank Prof. Mahoney for the constructive comments and valuable time. Detailed responses to these issues can be found behind the major comments.

Major comments:

1. Still no discussion of temperature dependence of viscous strain component

According to equation 10, the differences in ice temperature between the wet and dry experiments ought to result in an approximately two-fold difference in the viscous component of strain, $\varepsilon_v$. This should be discussed on more detail in relation to the overall increase in strain observed in the "floating" ice.

Re: The last paragraph of this paper discusses this. Please note that, although the differences in ice temperature between the floating and dry experiments appear to result in an approximately two-fold difference in the viscous component of strain, $\varepsilon v$, this does not mean that the temperature will make the viscous strain component of the floating ice only twice that of the dry ice. For example, the overall increase in strain for specimen Floating-5ppt-1 is much higher (fifteen times more) than that for specimen Dry-5ppt-1. This is mainly due to the aforementioned increase in effective dislocation density with increasing temperature (Cole and Durell, 2001; Cole, 2020). Corresponding discussion has been added in the revised manuscript (lines 402-408).

2. Ambiguous premise of paper

In their response to reviewer 1's comment 13, concerning the processes being tested by the experiment design, the authors state that "the original intention of this study is not to investigate the effect of temperature and floating condition on ice behavior" and the manuscript should instead be considered as a methods paper, as reflected by a minor restructuring of the conclusions section. I find this to be a disappointing climb-down, when I would have preferred to see a clarification of the conclusions that can be drawn from the authors' novel laboratory method.

Re: Earlier laboratory work on ice has mostly focused on dry, cold and isothermal ice. Here we wish to draw attention to the fact, that the results from such experiments may not apply directly to saline ice in its natural conditions, that is, when ice is floating in water – and also to the fact that the experiments where these conditions are mimicked in the laboratory can be performed. At this stage of the work we do not aim to study all of the factors that influence ice behavior. This point is now quite clear in the manuscript as explained in detail in our response to reviewer 1's comment.

Thus, in our response to reviewer 1's comment 13, we mean that the original intention of this study is not to study the effect of temperature on ice behavior (which usually requires a series of experiments with various ice temperatures) and not to explore the temperature-independent effect of water on ice behavior (which requires the ice to be tested under dry and wet conditions at a given ice temperature).

Our study does not fall into these two categories.

I also find this to be an entirely unconvincing argument. If the study was intended to be an investigation into the effects of temperature an floating conditions, why did the authors include experiments on both cold, isothermal non-floating ice and poly-thermal floating ice? Instead of removing references "wet" experiments, I urge the authors to revert to their nomenclature and instead spend a little extra text speculating on how floating, polythermal ice might respond differently to nonfloating ice with an equivalent average temperature.

Re: The manuscript has been modified in response to this comment (see lines 62-64). Although we regret the reviewer's disappointment, the fact is that the dry experiments with cold, isothermal non-floating ice were conducted to validate the performance of our newly developed testing apparatus and experimental methods, and the constitutive model under commonly used test conditions. Moreover, the dry, isothermal results provide a point of comparison for the floating ice results. The current nomenclature has satisfied reviewer 1 and is considered suitable by many experienced experts, which we have consulted.

In the manuscript, we have not speculated how floating, polythermal ice might respond differently to nonfloating ice with an equivalent average temperature.

In its current form, the premise of the manuscript is ambiguous and despite removing references to "wet" experiments, the text still contains phrases like "Thus, the fact that the ice was in contact with water and had a realistic temperature profile ..." (line 365), which still contribute to the misunderstanding that the authors are studying the effect of water on ice behavior.

Re: We have further modified the manuscript accordingly (lines 368 and 389), including the sentence that is considered misleading by the reviewer. Now it is obvious enough in the full text that this is not a type of study comparing the mechanical behavior of wet ice and dry ice at the same ice temperature (i.e., simply studying temperature-independent water effect).

[revised manuscript text omitted]